# Mechanical Properties and Flexural Response of Palm Shell Aggregate Lightweight Reinforced Concrete Beam

**Md. Habibur Rahman Sobuz** [1,*], **Md. Saiful Islam** [2], **Abu Sayed Mohammad Akid** [3], **Shuvo Dip Datta** [1], **Turki S. Alahmari** [4], **Noor Md. Sadiqul Hasan** [5], **Md. Tareq Hossain Khondoker** [1] **and Fahim Shahriyar Aditto** [1]

1   Department of Building Engineering and Construction Management, Khulna University of Engineering & Technology, Khulna 9203, Bangladesh; sd.datta@becm.kuet.ac.bd (S.D.D.); tareqkhondoker@becm.kuet.ac.bd (M.T.H.K.); fahim.aditto09@gmail.com (F.S.A.)
2   Department of Chemistry, Bangladesh Army University of Engineering and Technology, Natore 6431, Bangladesh; msaifuli2007@gmail.com
3   Department of Civil Engineering, Arkansas State University, Jonesboro, AR 72467, USA; abusayed.akid@smail.astate.edu
4   Faculty of Engineering, Department of Civil Engineering, University of Tabuk, Tabuk 47512, Saudi Arabia; talahmari@ut.edu.sa
5   Department of Civil Engineering, College of Engineering and Technology, International University of Business Agriculture and Technology, Dhaka 1230, Bangladesh; nmshasan@iubat.edu
*   Correspondence: habib@becm.kuet.ac.bd

**Abstract:** This work focuses on examining the mechanical characteristics and flexural response of reinforced concrete (RC) beams by incorporating oil palm shell (OPS) lightweight aggregate from oil palm shell waste. The OPS aggregates are replaced in various percentages, such as 0 to 50% of natural coarse aggregate (NCA). Mechanical properties of OPS concrete were conducted, and these properties were used to quantify the flexural performance of RC beams. Five RC beams with several gradations of OPS aggregates were cast and tested for this investigation. The first cracking, ultimate strength, load-deflection behavior, ductility index, and failure patterns of OPS aggregate beams were investigated as the corresponding behaviors to the NCA concrete beam. The fresh properties analysis demonstrated lessening the slump test by varied concentrations of OPS concrete. Furthermore, compressive strength was reduced by 44.73%, 50.83%, 53.33%, and 57.22% compared to 10%, 15%, 20%, and 50% OPC substitution at 28 days. Increasing OPS content in concrete mixes decreased splitting tensile strength, comparable to the compressive strength test. Modulus of rupture and modulus of elasticity experiments exhibited a similar trend toward reduction over the whole range of OPS concentrations (0–50%) in concrete. It was revealed that the flexural capacity of beams tends to decrease the strength with the increased proportion of OPS aggregate. Moreover, crack patterns and failure modes of beams are also emphasized in this paper for the variation of OPS replacement in the NCA. The OPS aggregate RC beam's test results have great potential to be implemented in low-cost civil infrastructures.

**Keywords:** oil palm shell; natural coarse aggregate; flexural response; ultimate capacity; mechanical properties

## 1. Introduction

In the past decades, a significant increase in palm oil production has been introduced in the oil palm industry in Malaysia. The oil palm agronomy zone was expanded to approximately 5.81 million hectares, and the palm trees fresh fruit bunch (FFB) yielded around 17.89 tons/hectare [1,2]. Furthermore, the palm oil industry followed FFB processing and obtained more than 19.92 million tons of palm oil per year [2]. Many environmental issues have arisen due to the building industry's fast global expansion due to the overuse of natural resources [3]. Due to large production, remaining solid scum and liquid wastes are

produced from the oil palm manufacturing process. According to Sukiran et al. [4], more than 4 million tons of solid waste were produced from Malaysia's oil palm shell. These massive volumes of solid waste ingredients are frequently generated in the mills, making storage difficult. Usually, the unoccupied fruit bunches are burned and create severe air contamination. As a result, an expensive situation arises to dispose of these residues while achieving environmental guidelines. In this circumstance, several efforts to develop these by-products are going in full stream by adding extra value to the products. Consequently, the application of oil palm shell (OPS) wastes as a substitution for natural coarse aggregate (NCA) in building materials has arisen as an attractive solution instead of disposing of it on the land.

Numerous perspectives have been established to explore the properties of concrete incorporating various categories of waste materials [5–10]. Some researchers investigated using palm shell wastes to produce lightweight concrete (LWC) [11–13]. OPS are the stony hard endocarp, but they are not heavy and are the correct size. Once bound in the concrete matrix, the organic surfaces are too rigid to contaminate or leach to form harmful chemicals. Lightweight OPS concrete can be achieved because OPS is less dense than the regular coarse aggregate. In addition, Olanipekun et al. [14] used palm kernel and coconut shell waste materials as suitable size aggregates in different proportions and then compared the properties among them. Though OPS is a hard endocarp and stony in nature, it is lightweight and generally suitable for use because the hard surfaces from organic sources are not contaminated to generate toxic elements when used in the concrete matrix. Consequently, it can be utilized to produce lightweight concrete as a decent replacement for NCA [15,16]. When coarse aggregate is replaced with coconut shell, the density drops substantially, requiring a higher cement quantity to retain the optimal compressive strength of concrete specified in the design. Furthermore, it was obtained that concrete containing OPS aggregate as a replacement for stone chips has a low density that fluctuates from 1700 to 2050 kg/m$^3$ in the range [15].

Among all properties of OPS concrete, the main concerning factor is to gain compressive strength. Previous researchers reported that the development of the compressive behavior of OPS concrete largely depends on the quality of the OPS fruit bunches and the mix preparation method [15]. OPS concrete easily achieves more than 17 MPa compressive strength, which is a requisite for LWC structure as per the standard practice of ASTM C330 [17]. A study on OPS concrete reported that 28 MPa compressive strength could be obtained by partially substituting the NCA with OPS [11]. It was also reported that palm shell concrete specimen failure behavior is usually directed by the quality of OPS [18]. In addition, the mechanical characteristics of concrete with OPS have been investigated by several researchers to determine the efficiency of OPS as aggregate [12,19–21]. However, Teo et al. [22] reported that OPS might be implemented instead of aggregates to generate the LWC structure. The mechanical behaviors of concrete with palm kernel shell (PKS) were compared with orthodox concrete that added mineral admixtures [23,24]. Hence, the inclusion of admixtures in oil PKS concrete improved the hardened phenomenon, including compressive strength and suitable elastic modulus. The impact of Totally Accumulated Weld Volumes (TAWV) and various groove inclinations (45°, 60°, 75°, and 90°) on the elastic properties of AISI 8620 and AISI 1040 cylindrical steel joints were studied experimentally by Adin [25]. In addition, Maghfouri et al. [18] determined the optimum OPS content to replace stone chip particles in concrete. As a result, they concluded that OPS particle concentration might be within 60% of the total replacement of NCA based on mechanical properties.

Numerous investigations have been carried out to examine the performance of lightweight reinforced concrete beams incorporating various materials as coarse aggregate replacement [26–29]. In addition, Olanitori and Okusami [30] reported the effect of shear rebar on the flexural behavior of PKS and control concrete beams. They concluded that the beams containing 20% partial replacement of PKS could be cast off for low-cost structural purposes. Alengaram et al. [13] studied the ductility response of RC beams with varying

substitutions of PKS. They discovered that the displacement of ductility for the NSC and PKS beams was in the range of 2.5 to 3.0 and 4.0 to 6.0, respectively. The mode of failure of PKS beams exhibited ductile behavior, whereas the NSC beams demonstrated a relatively brittle failure nature under transverse action of loading. In another research effort, the flexural performance of RC beams made with the varying gradation of OPS aggregates was conducted by Teo et al. [11]. The study suggested that the LWC of OPS aggregate could be replaced fully when the special mix design is considered to achieve the desired structural performance. However, partially replaced palm shell has the potential to be used in construction industries. According to Shafigh et al. [26], it was possible to obtain ductility response from OPS aggregate concrete, which served as an environmentally friendly LWC compared to artificial LWC. Yap et al. [31] compared the flexural and ductility performances between OPS concrete beams and steel fiber RC beams. They reported that steel fiber leads to the enhancement of the beams' mechanical behavior and moment capacities. The steel fibers significantly improved the ultimate load in the specimens compared to OPSC beams; however, the maximum deflection was reduced in OPSC beams with steel fibers.

The primary output of this paper is to examine the flexural responses of an RC beam with varying concentrations of OPS aggregate as a replacement for the conventional natural aggregate. The intention was to investigate the practical application and performance of OPS lightweight aggregate in reinforced concrete beams to explore the use of waste products for adding value in the efforts of the construction industry towards achieving sustainability goals. The flexural response of RC beams and the utilization of the optimum percentage of a lightweight aggregate in the structure added significant value to Malaysia's construction industry. Furthermore, this study makes distinct contributions and introduces novel elements by exploring the flexural responses of reinforced concrete (RC) beams with different oil palm shell (OPS) aggregate concentrations. Additionally, it examines the mechanical properties of OPS aggregate concrete and compares the failure patterns and crack behaviors with those of normal strength concrete (NSC) beams. This paper then starts by determining the mechanical properties of OPS aggregate concrete. These properties are then used in a rectangular stress block analysis to predict the flexural response of the OPS concrete beam at the cracking, ultimate limit, and ductility behavior of RC beams with an analytical approach. Furthermore, the failure pattern and crack behaviors were observed for varying OPS-reinforced concrete beam concentrations and the corresponding behaviors were compared to the NSC beam.

## 2. Materials and Methods

### 2.1. Materials Selection and Mix Composition

For this investigation, the concrete mix compositions were fixed and represented the ratio of 1:1.65:2.45 with the content of Ordinary Portland cement, river-washed sand, and maximum crushed coarse aggregate of 20 mm in size to obtain the desired strength of lightweight concrete. The OPS was replaced with crushed natural aggregate in the weight of cement at 0%, 10%, 15%, 20%, and 50%, and the $w/c$ of 0.45 was kept constant for each mix. The physical characteristics of OPS aggregate vary from the crushed natural aggregates due to its organic nature, and the average size range of OPS aggregate particles varies between 12.5 mm and 9.5 mm. Figure 1 represents the fresh fruit branch with OPS configurations before the palm oil extraction in the milling process. Figure 2 highlights the working methodology of the entire research project. The physical characteristics of crushed natural aggregate and OPS are illustrated in Table 1. In addition, the laboratory's reinforcing steel properties were determined by experimental testing, as illustrated in Table 2.

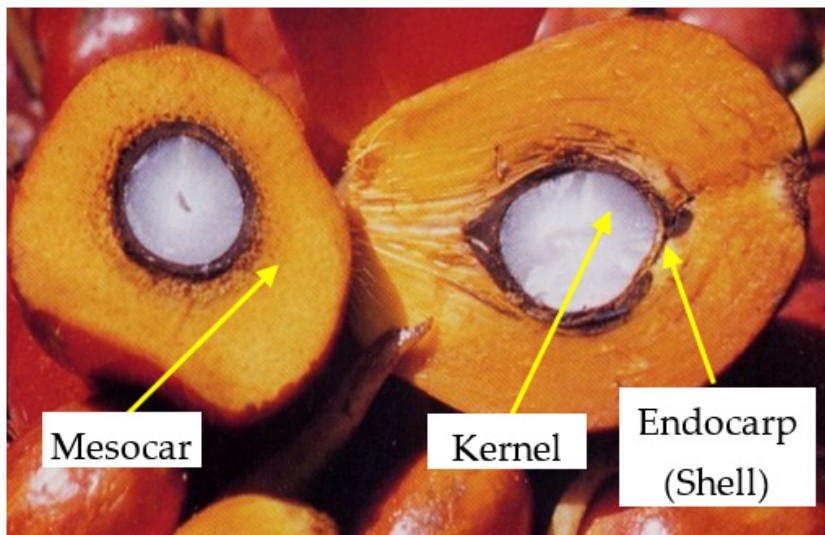

**Figure 1.** FFB and OPS configurations before the oil extraction.

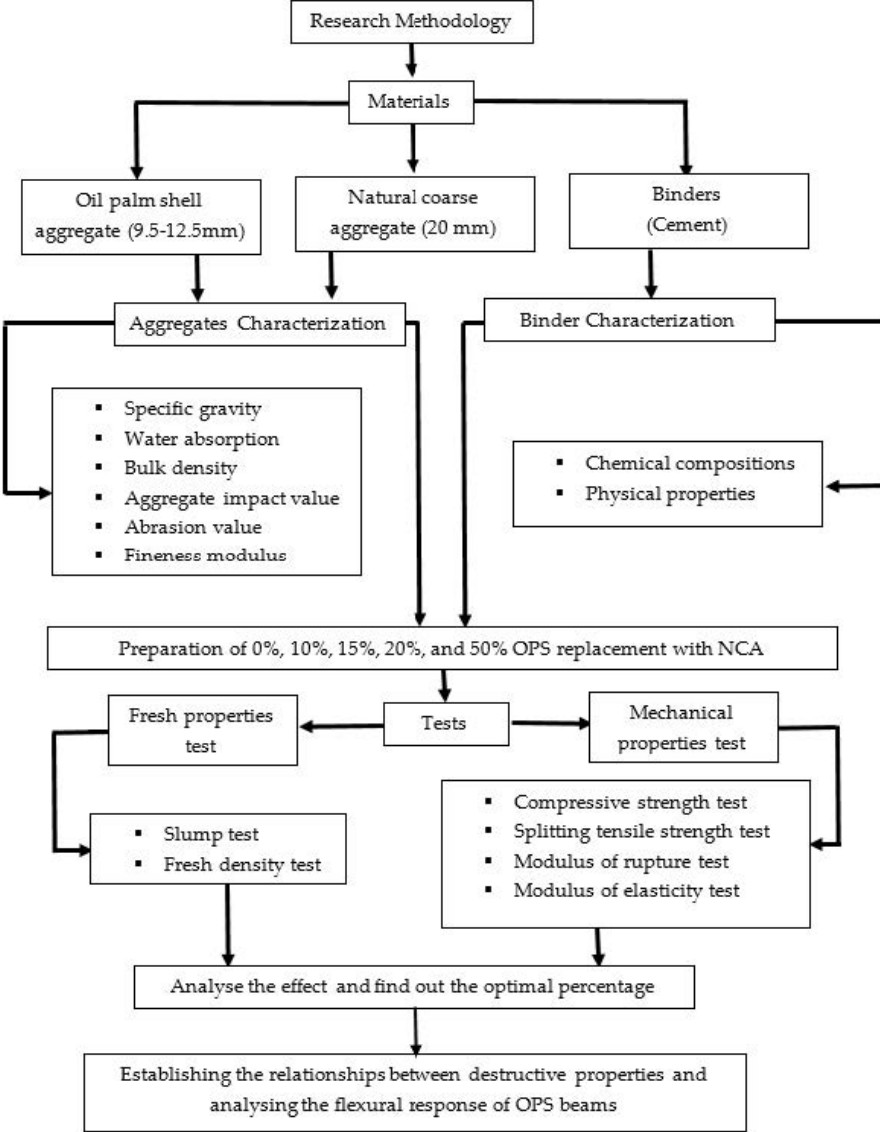

**Figure 2.** Working methodology of this research.

**Table 1.** Physical properties of palm shell and natural aggregate.

| Properties | Palm Shell Aggregate | Natural Aggregate |
|---|---|---|
| Bulk density (kg/m$^3$) | 572 | 1445 |
| Water absorption for 24 h (%) | 25.64 | 0.7 |
| Specific gravity | 1.21 | 2.72 |
| Aggregate impact value | 6.65 | 12.32 |
| Los Angeles abrasion value (%) | 5.1 | 24.5 |
| Aggregate crushing value | 6.78 | 17.92 |
| Thickness of shell (mm) | 0.5–4.0 | 5–20 |
| Fineness Modulus | 6.24 | 6.76 |
| Maximum aggregate size (mm) | 12.5 | 20 |

**Table 2.** Properties of reinforcing steel.

| Rebar Type | Yield Capacity (MPa) | Elastic Modulus (GPa) |
|---|---|---|
| Compression, T10 | 482 | 195 |
| Tension, T10 | 482 | 195 |
| Compression, T6 | 470 | 189 |
| Shear stirrup, R6 | 215 | 200 |

*2.2. Specimen Configuration, Preparation, and Curing*

In this study, fresh and mechanical concrete properties, including varying OPS dosage, were designated with the symbol "PSC-C" using the subsequent process. The primary letter "PSC" denotes the palm shell concrete; another lettering, "C" indicates the OPS dosage as a percentage replacement in the conventional natural aggregate. Furthermore, five RC beams with various OPS gradations were considered for this current investigation. The first beam was marked without replacing the OPS aggregate as designated OPSB-0 (control beam). The rest of the four beams were designated as OPSB-10, OPSB-15, OPSB-20, and OPSB-50, and the suffix to the beam designation indicates OPS aggregate replacement densities in the NCA. Table 3 shows the test program to assess the flexural response of RC lightweight OPS aggregate beams.

**Table 3.** Test program.

| OPS Replacement Level (%) | 0 | 10 | 15 | 20 | 50 |
|---|---|---|---|---|---|
| Beam Designation | OPSB-0 | OPSB-10 | OPSB-15 | OPSB-20 | OPSB-50 |

The longitudinal and X-sectional configuration of the RC beams were prepared with and without OPS aggregate, as illustrated in Figure 3. All of the beams have a width of 150 mm, a depth of 150 mm, and a span of 1500 mm. All RC beams were longitudinally reinforced with the flexural tensile bar designated as 2T-10 bars (two 10 mm diameter bars). The 2T-10 bars for the beams were labeled with OPSB-50 as compression bars; consequently, the beams designated with OPSB-0, OPSB-10, OPSB-15, and OPSB-20 were used for the compression reinforcement of 2T-6 bars (two 6 mm diameter bars). Furthermore, the shear reinforcement of R-6 stirrups (bar diameter = 6 mm) was located at an interval of 100 mm to 125 mm along the beam span to resist the shear failure. All RC beams were concealed with plastic sheets after the casting period of 2 h and kept for 24 h in the laboratory. The formwork was removed after 24 h; therefore, the beam curing process was conducted by the wet hessian wrapped around the beam and spraying water regularly before the test day

in the laboratory. The relative was in the range of 82–87%, and the temperature difference was approximately $30 \pm 2\,^{\circ}\text{C}$.

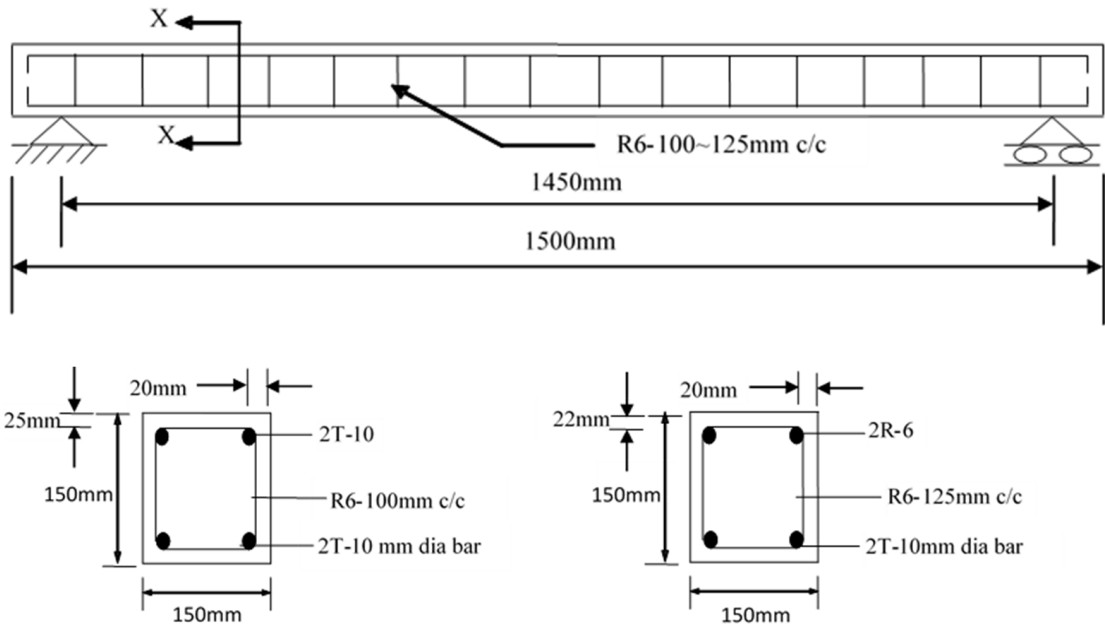

Section X – X for beam OPSB-50

Section X – X for OPSB-0, OPSB-10, OPSB-15 and OPSB-20

**Figure 3.** Longitudinal and cross-section (X-X) details of the investigational beams.

### 2.3. Instrumentation and Testing Procedure

#### 2.3.1. Fresh Properties

In the present experiment, the slump test was carried out in accordance with ASTM-C143/C143M-20 [32]. BS EN 12350-6 [33] was used to conduct the fresh density test. Figure 4 depicts the testing procedure of fresh properties.

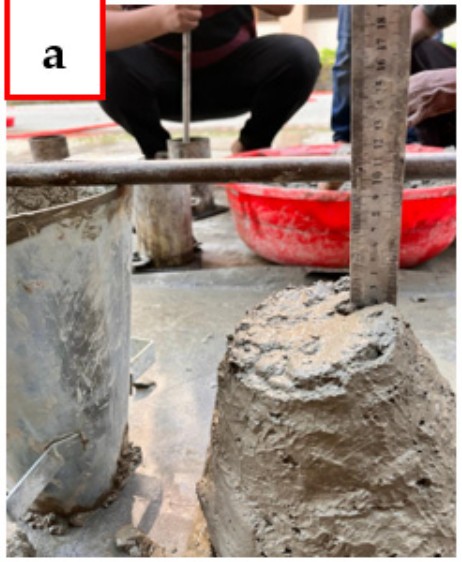
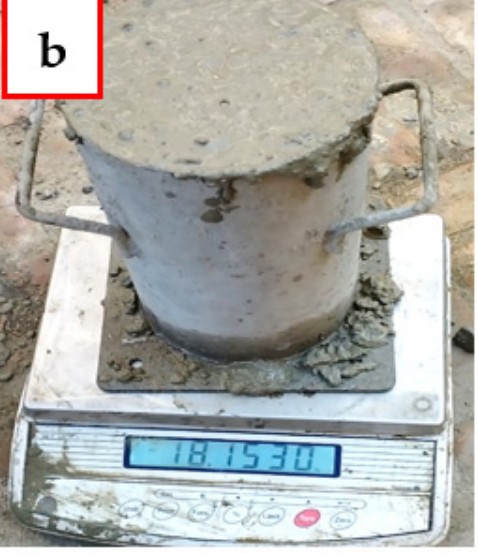

**Figure 4.** Testing procedures of (**a**) slump and (**b**) fresh density test.

### 2.3.2. Mechanical Properties

Standard samples were tested to quantify the mechanical properties of varying densities OPS aggregate mixes that consisted of the cube, splitting tensile, and modulus of rupture strength at 7- and 28-day curing specimens, whereas modulus of elasticity was carried out at 28 days. An electronic digital compressive testing instrument with a capacity of 3000 kN was regulated to determine both compressive and splitting tensile strengths, whereas a 150 kN capacity digital flexural testing machine was employed for flexural strength. The uniaxial compressive test was carried out with complying BS EN 12390-3: 2002, where the dimension of the cylinder was 100 × 200 mm for testing [34]. In addition, the tensile strength test by splitting the specimen was performed with complying ASTM C496/C 496M–17 (2017) using a 100 mm × 200 mm cylinder [35]. Prism specimens comprising 100 mm × 100 mm cross-sections and span lengths of 500 mm were investigated for flexural strength based on the standard ASTM C78/C78M-18 [36]. The elastic modulus property of concrete mix was examined based on the standard ASTM C469/C469M-14 [37] using a cylindrical specimen of 150 × 300 mm under compression. Figure 5 represents the mechanical properties testing procedures.

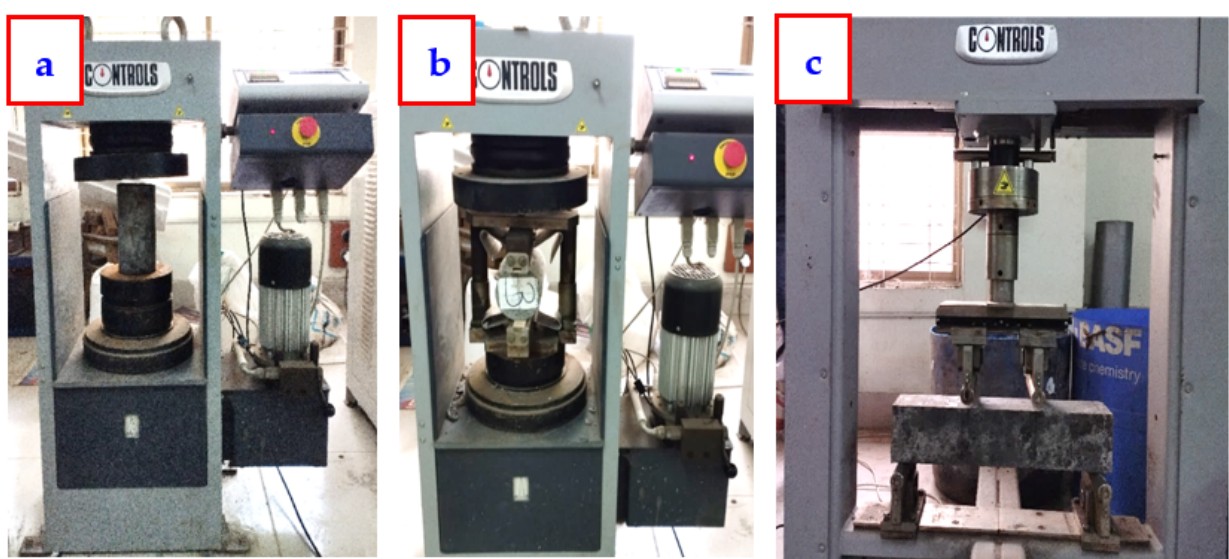

**Figure 5.** Test setup of (**a**) compressive, (**b**) splitting tensile, and (**c**) flexural strength test.

### 2.3.3. Flexural Testing of Beam

The flexural testing scheme of the OPS RC beam was arranged in the laboratory with an effective span length of 1450 mm, as illustrated in Figure 6. To determine the vertical contraction of the beam, linear voltage displacement transducers (LVDTs) were positioned at the tension soffit of the beam, including two at the quarter distance of span and one at the mid-span. The monotonic concentric load was applied using a spreader steel I section to the beam through the hydraulic jack attached to a load cell. In addition, the LVDTs were associated with the portable data logger with the data acquisition process. The values of deflection and load were recorded at regular intervals with the increase in load to evaluate the flexural response of the OPS beam. Crack width and propagation were visually observed and determined using a magnifying glass with a scale of 40 and a sensitivity value of 0.01 mm, along with the recorded load value.

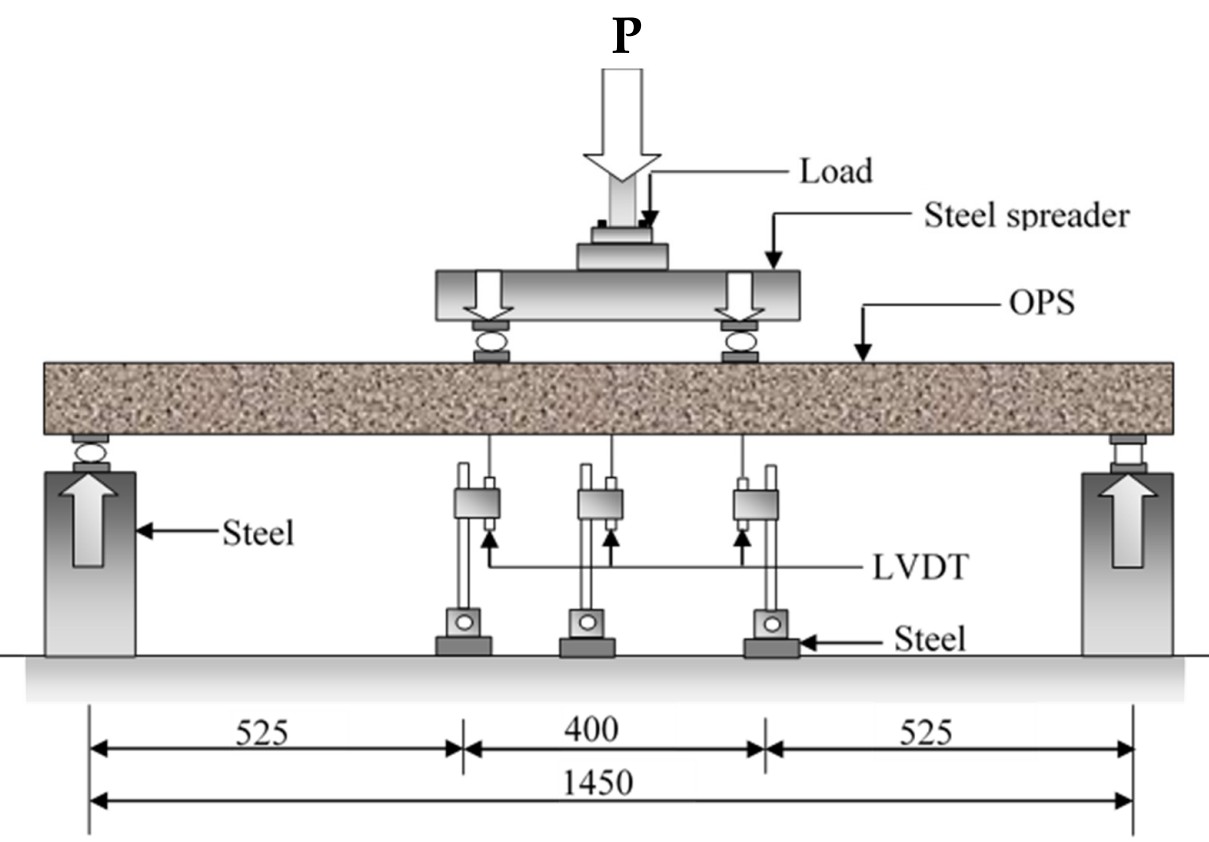

**Figure 6.** Loading and instrumentation of flexural testing of OPS beam in the laboratory.

*2.4. Analytical Methods*

2.4.1. Flexural Response of RC Beam

The analytical approach can predict the flexural behavior of the RC beam element, which is briefly described by the BS 8110-1 code to simulate the first crack behavior and ultimate load-carrying capacities [38]. Therefore, a rectangular stress block method is applied with the BS standard to find the equilibrium forces, including concrete compression, steel reinforcement at tension, and compression, which are performed on the reinforced concrete (RC) beam element. According to the codes, the traditional sectional analysis was adopted for strain compatibility, denoted as "plane sections remain plane", and the stress vs. strain interactions of reinforcement and concrete are utilized for equilibrium equations of the forces, as illustrated in Figure 7.

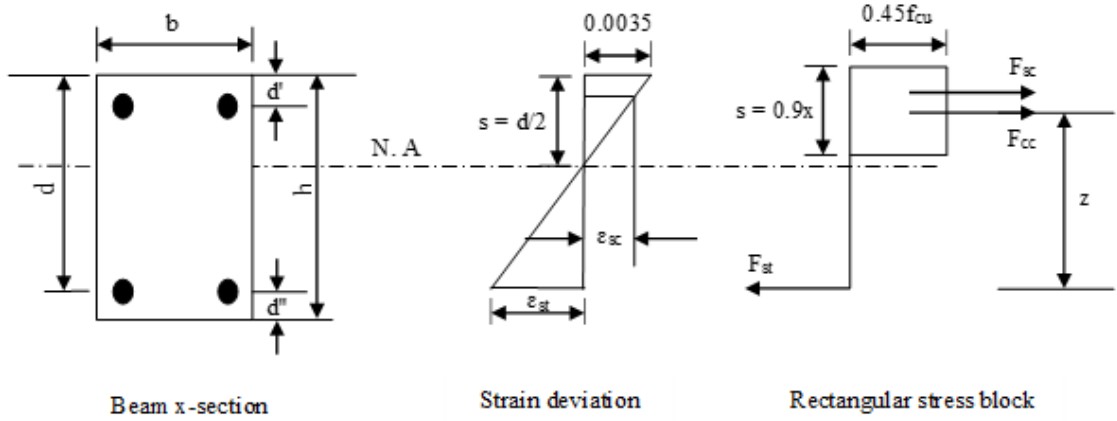

**Figure 7.** Predicting the flexural strength of RC beam with rectangular stress block analysis.

Therefore, the first cracking moment of the RC beam ($M_{cr}$) is obtained by a theoretical formula as represented below:

$$M_{cr} = \frac{f_r I_g}{y_t} \tag{1}$$

Here, it is considered as the arm between the soffit of the beam and the neutral axis, indicating the second moment of inertia about the neutral axis of beam x-section, and $f_r$ is considered the flexural strength of the concrete.

For the equilibrium of the section from the stress-strain diagram (refer to Figure 2), it can apply the following equations:

$$F_{st} = F_{cc} + F_{sc}$$
$$\to 0.95 A_s f_y = 0.45 f_c' bs + 0.95 A_s f_y \tag{2}$$

The tension and compression steel strain can be measured by using Equations (3) and (4).

$$\frac{0.0035}{x} = \frac{\varepsilon_{st}}{d-x}$$
$$\to \varepsilon_{st} = \left(\frac{d-x}{x}\right) \times 0.0035 \tag{3}$$

Similarly,

$$\varepsilon_{sc} = \left(\frac{x-d'}{x}\right) \times 0.0035 \tag{4}$$

The relative stress and force in the compression zone steel ($f_{sc}$) is

$$f_{sc} = \varepsilon_{sc} \times E_{sc}$$
$$\to f_{sc} = \left(\frac{x-d'}{x}\right) \times 0.0035 \times E_{sc} \tag{5}$$

Then,

$$F_{sc} = \left(\frac{x-d'}{x}\right) \times 0.0035 \times E_{sc} \times A_{sc} \tag{6}$$

We can take the moment at the top of the lever arm (refer to Figure 2).

$$M_{ult} = F_{st} \times d - F_{sc} \times d' - F_{cc} \times 0.45 \times x \tag{7}$$

2.4.2. Ductility Index

Ductility is a significant characteristic of any infrastructure component. While the ductility term is used for the RC members, it indicates the capability to withstand substantial inelastic displacement earlier to collapse. In RC members, ductility is typically referred to as the ductility index/ratio. This study measures the displacement ductility index, representing the displacement ratio at the failure stage and the yield point displacement. Therefore, the displacement ductility index can be obtained by implementing the following equation:

$$\mu_{d\delta} = \frac{\delta_u}{\delta_y} \tag{8}$$

where ductility index due to deflection, $\delta_u$ = ultimate deflection at mid-span, and $\delta_y$ = mid-span deflection at the yields of the bottom bar.

**3. Results and Discussion**

*3.1. Properties of Fresh Concrete*

Figure 8 graphically represents the slump and density of different OPS percentage replacements in NCA for the concrete mixtures. The slump of the concrete mix linearly reduced with an increasing percentage of OPS in concrete. The range of slump values with different concentrations of OPS aggregate concrete mixes was 62–29 mm, indicating an inferior degree of workability than control concrete. The reason for lower workability is

attributed to the higher capacity of water absorption rate and porosity nature of OPS aggregate because of the mortar attached to it. The test results look reliable; other researchers also observed related findings [39–41].

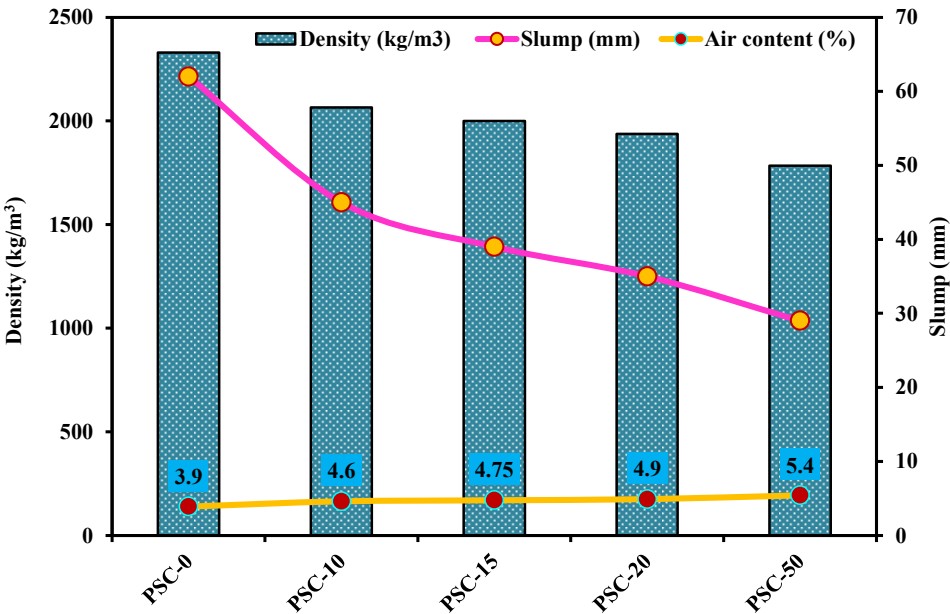

**Figure 8.** PSC concrete mix for slump, density and air content.

It can be seen from Figure 8 that the values of concrete density differed between 2330 kg/m$^3$ to 1784 kg/m$^3$ which was related to the upsurge in OPS concentration level. It seems to minimize the dead load of the RC members, which satisfied the LWC requirement according to ASTM C330/C330M–17a [18]. Other researchers also reported a similar finding in their experimental investigation on structural lightweight concrete production with expanded clay [42]. However, LWC is considered with a density below 2160 kg/m$^3$ based on a practical application [43]. It can be highlighted that the decline in density was noticed for OPS lightweight concrete, extending from 11% to 23% of the NCA concrete. Shafigh et al. [42] observed this consistent phenomenon of test results for palm shell LWC, where the density was approximately 14–19% lighter than NCA concrete. The decrease in the density of palm shell aggregate lightweight concrete is observed due to the more permeable characteristics of the palm shell that is stuck to the aggregates. In addition, varying gradations of sand and coarse aggregate used in different zones and aggregate interlocking nature are given the highest inhibition that may significantly influence its density. Moreover, this gradual loss of density can be attributed to the introduction of a higher percentage of void present due to the addition of OPS in natural aggregate, especially at higher replacements of OPS concentration. Finally, the above investigations exhibited that OPS concrete would be considered a relatively low-cost building material as a secondary coarse aggregate in terms of reducing strength/economy ratio, environmental impact, and ultimately achieving sustainability goals [14,44–46]. It should be noteworthy from Figure 8 that there is a noticeable decreasing trend in the slump with the increasing of the OPS concentration replacement level in the natural aggregate; however, the air content amount in the OPS concrete exhibited a pattern of increasing linearly which was more than the conventional mix.

Furthermore, the air content of OPS concrete mixtures has been obtained in the range between 3.9% and 5.4% for varying concentrations of OPS replacement in natural aggregate. The experimental investigation by Teo et al. [22] concluded that the replacement by OPS in concrete exhibited air content ranges of 4.8 to 5.5%, which is closely similar to this study. In addition, the OPS concrete mix designated as PSC-50 with the highest replacement of palm shell had demonstrated a notably lowest slump; in contrast, the PSC-50 mix increased

the air content by around 39% which was more than the conventional mix PSC-0. This higher OPS aggregate air content percentage was observed due to the very irregular shape obtained from the factory, which resisted achieving full compaction and OPS particle's high water absorption properties. Consequently, the slump with a first OPS replacement of concrete PSC-10 was 27% lower than the natural aggregate concrete; in contrast, the air content of the PSC-10 was 18% which was more than the concrete mixed without OPS.

### 3.2. Mechanical Characteristics of Concrete

#### 3.2.1. Compressive Strength

The test results of the compressive characteristics of strength with different OPS densities in the substitution of coarse aggregate are shown in Figure 9 for 7- and 28-day curing ages. In this study, OPS aggregate lightweight concrete mixes have attained about 79–83% of the 28-day compressive strength at 7 days of curing. In addition, a similar consistent trend of the 7- and 28-day strength ratio of the test result for OPS concrete was found in the previous work [39,41]. The laboratory experimental results show that the increment in OPS concentration in concrete mix reduces the compressive strength at 7- and 28-days. It was obtained at 44–45% for PSC-10, 49–51% for PSC-15, 51–54% for PSC-20, and 55–57% for PSC-50 over the natural concrete mix PSC-0. Overall, the OPS aggregate concrete specimens showed lower strength for 7- and 28-days than the conventional concrete mix. Khankhaje et al. [46] concluded that the decreasing fashion in compressive strength was demonstrated by adding oil palm kernel shell (PKS) aggregate in concrete mixes, especially at the upper replacement level. OPS concrete mixes usually have demonstrated lower strength due to compression in the limit of 42–55% at 28 days compared to the conventional concrete mix [47]. Furthermore, the consistent finding by Mannan et al. [12] stated that concretes having OPS aggregate showed about 52% less compressive strength than conventional aggregate mix concrete. Numerous research efforts have also revealed that the response of compressive strength, including OPS aggregate, is lower grade as they produced approximately 25 MPa concrete [12,14,48]. Hence, it could be said that adding OPS aggregate in the regular concrete mix leads to a decrease of almost half of the strength of the control mix. The reduction in strength could be attributed to the growth of small hairline cracks in the specimens, particularly at higher replacement levels of OPS. Moreover, poor bonding of the interfacial transition stage and cement paste does not compensate for the gaps between fine aggregate and the varying irregular shape of OPS aggregates.

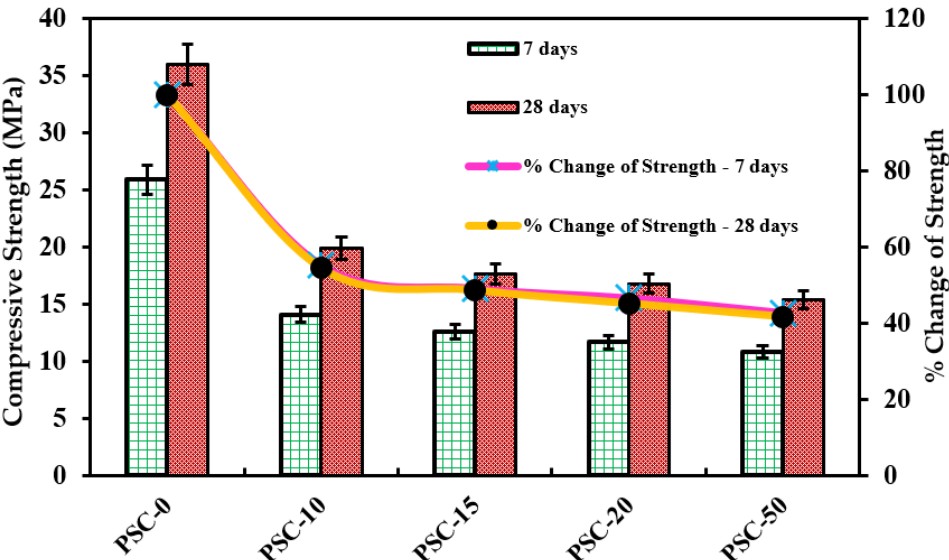

**Figure 9.** Compressive strength values for varying OPS concrete mixes.

Notably, the maximum reduction in strength in compression testing was observed for the PSC-10 mix for the control one PSC-0 at 28 days, as shown in Figure 6; consequently, a minor decline in the compressive strength was demonstrated for the subsequent concrete mixes with varying OPS replacement. The current experimental investigation revealed that OPS concrete compressive strength at day 28 and the fresh density of the concrete mixes range from 15.4 to 36 MPa and 1784 to 2330 kg/m$^3$, respectively, which are satisfied by the code of standard requirement for lightweight concrete production [49]. As expected, it was seen that the OPS aggregate concrete manufactured in this current investigation is less heavy than the NCA concrete, which varied from 24% to 12%. These test results provided quite similar agreement with the investigation conducted by Chen and Liu [50] on lightweight aggregate concrete and obtained a fresh density around 36% smaller than the control concrete. This lighter nature of OPS aggregate concrete was also observed due to the different gradations of OPS and fine aggregate used in different regions, air-entraining, the weak bond between cement composite and OPS, and the interlocking of natural aggregate and OPS. Finally, it could be concluded that OPS aggregate use in concrete leads to a substantial saving of the dead load of the concrete structures.

3.2.2. Splitting Tensile Strength

The splitting tensile strength for all the concrete mixes at different palm shell aggregate concentrations and conventional aggregate at the 7- and 28-day curing ages is illustrated in Figure 10a. It is noteworthy from Figure 10a that as the different concentrations of OPS aggregate amplified in the concrete mixtures, the tensile strength gradually reduced for the 7- and 28-day curing ages. It was revealed from the test results that the different concentrations of OPS replacement in concrete at 7 days have attained the customary range of 79–83% of its 28-day tensile strength through the split tensile test of the specimens. It is also obtained from Figure 10a that the tensile strength value is more than 1.3 MPa and 2.0 MPa at the 7-day and 28-day curing ages, respectively, for all OPS concrete mixes. It satisfied the minimum value required for lightweight structural concrete while complying with ASTM C330 [17]. The tensile strength directly exploited a similar decreasing fashion in the compressive strength results when increasing the OPS replacement in the concrete mixtures. The test results are consistent with Mannan and Ganapathy [31], who obtained the splitting tensile strength at 7 and 28 days of OPS aggregate concrete, which varied from 1.35 MPa to 1.78 MPa, respectively. In addition, the decreasing rate of tensile value of concrete with varying concentrations of OPS at 28 days was to the extent of 15.3–28.8% compared to the conventional coarse aggregate mix PSC-0. The test results of OPS concrete carried out by Shafigh et al. [51] and this study revealed that the inclusion of OPS aggregate concrete exhibited a consistent decreasing trend of splitting tensile strength. This result implies that even with using the different percentages of OPS in the concrete, the decrease in the splitting strength using palm shell aggregate is not much greater than the control one.

The split-to-compressive strength ratio of palm shell aggregate lightweight concrete at 7- and 28-day test specimens is in the average range of 8–14%. In addition, the correlation ratio of split to compressive strength with the corresponding compressive results of palm shell aggregate lightweight concrete is shown in Figure 10b with the strong coefficient of regression $R^2 = 0.96$ and $R^2 = 0.99$ at 7- and 28-day, respectively. This study observed that the tensile behavior of the palm shell aggregate lightweight specimen is about 8–14% of the compressive strength [14], which strongly agrees with this current investigation. Hence, lightweight concrete made with OPS aggregate is still to be recognized as better quality concrete according to the proportion of split to compressive strength compared to conventional coarse aggregate concrete.

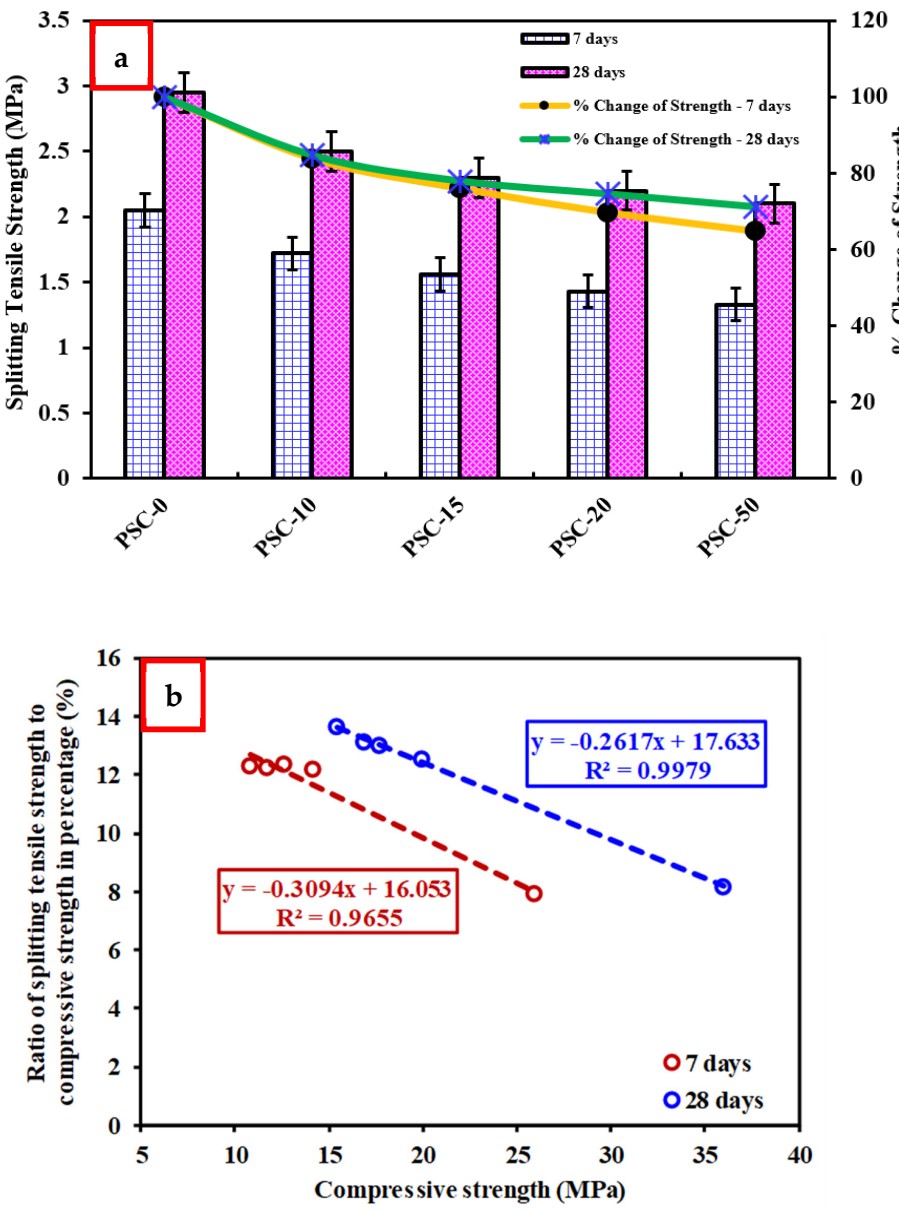

**Figure 10.** (**a**) Splitting tensile strength values for varying OPS concrete mixes. (**b**) Correlation between the ratio of splitting tensile strength to compressive strength and compressive strength for varying OPS concrete mixes.

### 3.2.3. Modulus of Rupture Strength

The experimental outcomes for the modulus of rupture characteristics of concrete, including varying OPS aggregate concentrations for 7 and 28 days, are presented in Figure 11a. The modulus of rupture strength of concrete without OPS replacement (control) was recorded at 2.56 MPa and 3.7 MPa for 7 and 28 days, respectively. Results show that the varying concentration of OPS addition in concrete decreased the modulus of rupture strength for both 7- and 28-day testing specimens. Furthermore, the early age of 7 days modulus of rupture strengths of OPS concrete was in the series of 1.83–1.56 MPa, which is in the extent of 29 to 39% less than the control one; albeit for the age of 28 days, 2.6–2.4 MPa was obtained with the extent of reduced modulus of rupture strengths of about 30 to 35% compared to the control one. In addition, this reduction range of rupture strength modulus is consistent with the conclusions of previous experimental investigations for OPS aggregate lightweight concrete [20,52]. The reason behind such behavior of modulus of rupture strength of OPS aggregate concrete could be explained as follows: less packing

aptitude of the fine aggregate and the empty space within the cement paste could help increase the void development between the OPS particles and the cement composite.

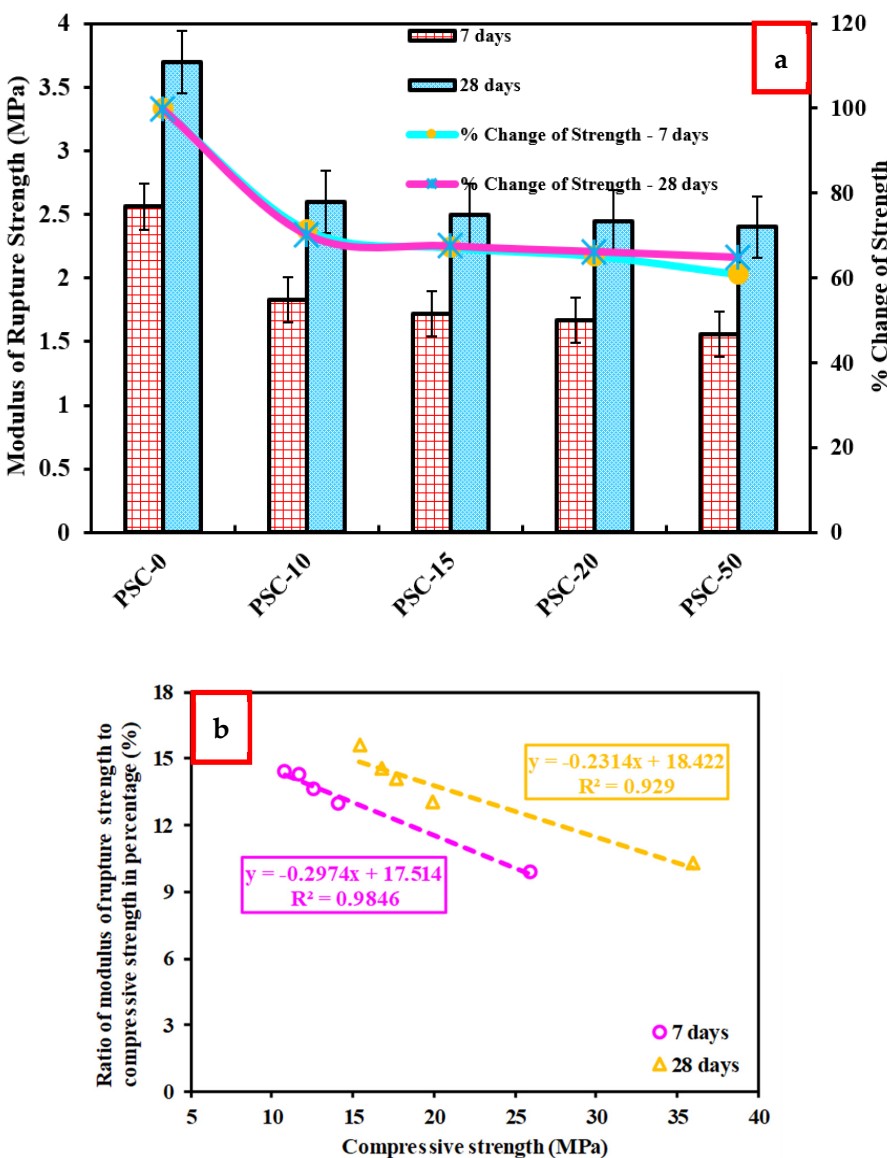

**Figure 11.** (**a**) Variation of modulus of rupture strength with varying concentrations of OPS mixes. (**b**) Correlation between the ratio of flexural strength to compressive strength and compressive strength for varying OPS concrete mixes.

Regarding the control mix of PSC-0 at 7 days, it has achieved approximately 69% modulus of rupture strength of its 28-day strength. In addition, increasing the OPS concentration in concrete mixes at 7-day curing strength was examined, which varied from 65–to 70% of its 28-day curing strength. It expressed the considerable modulus of rupture strength at early ages for lightweight structural concrete. It was also obtained that adding more than 10% OPS concentration resulted in a lower modulus of rupture strength, which might be the reason for the increase in matrix porosity due to the generation of more voids.

It can be seen that the experimental results of the modulus of rupture strength of all OPS concrete specimens are incorporated into the analytical investigation to calculate the flexural response of RC LWC beams. It was also obtained that the modulus of rupture strength is about 10.3% on average of the control mix's compressive strength, which consistently agrees with the test results where the ratio was obtained around 11% at 28 days [53].

The experimental investigation revealed that the modulus of rupture strength of varying concentrations of OPS concrete is in the limit of 2.6–2.4 MPa; whereas compressive strength varies from 19.9–15.4 MPa, and the modulus of rupture strength to compressive behavior ratio is varied between 13 to 16% at 28 days [54].

Moreover, the correlation ratio of modulus of rupture to compressive behavior with the corresponding compressive strength of OPS concrete is shown in Figure 11b with the best regression coefficient $R^2 = 0.98$ and $R^2 = 0.92$ at 7 and 28 days, respectively. Further, the exploratory results, the splitting, and the modulus of rupture strength ratio for OPS aggregate concrete of PSC-10, PSC-15, PSC-20, and PSC-50 mixes were obtained at 96%, 92%, 89%, and 87%, respectively, in this current investigation. These percentages indicate that the ratio decreases with the increase in OPS concentration in the mixes and comparatively higher splitting tensile strength gain than the modulus of rupture strength. The lower splitting/modulus of rupture strength ratio for OPS aggregate concrete was obtained in the 65% to 75% range, which was also observed by Alengaram et al. [55] in their experimental investigation.

### 3.2.4. Modulus of Elasticity

Figure 12 illustrates the graphical plot of the modulus of elasticity (MoE) of concrete mixes with varying OPS concentration levels as a replacement for conventional coarse aggregate at 28-day test samples. The MoE of concrete is generally influenced by a number of features, such as compressive behavior, density, and type of aggregate in the mixes, the interfacial bond of the aggregate and cement composites, and the elastic properties of the materials composition utilized in the concrete mixes. The MoE was obtained at 13.4–28.6 GPa for the OPS replacement concentration level of 50–0%, which the FIP manual satisfied for the MoE of lightweight structural concrete. A similar consistent MoE of OPS lightweight concrete was observed by [56], which presented the MoE range of lightweight concrete between 15.2 and 14.5 GPa. Figure 9 shows that the MoE of PSC-10, PSC-15, PSC-20, and PSC-50 mixes decreased by about 40%, 43%, 47%, and 53%, respectively, to the mix PSC-0 for the similar trend observed for the compressive, split, and modulus of rupture strength test of this study. This reduction trend of MoE could be attributed to the flat peripheral outward of the palm shell aggregate with both concave and convex advents, confirming the weak interfacial transition zone of the palm shell aggregate and cement matrix. The finding aligns with the test outcomes conducted by Shafigh et al. [26], who observed that weak bonding of OPS aggregate and internal cement paste results in a lower modulus of elasticity.

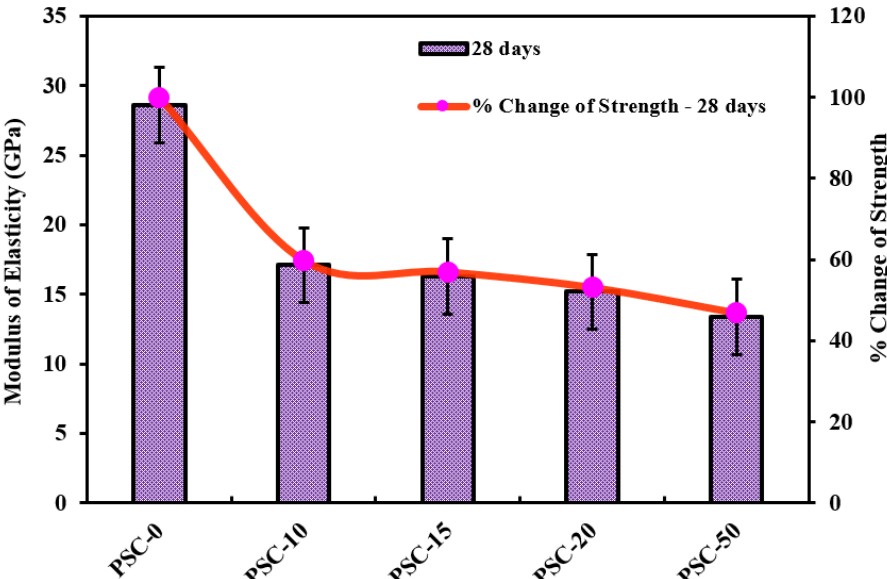

**Figure 12.** Variation of modulus of elasticity with varying concentrations of OPS mixes.

### 3.3. Comparison between Experimental Results and Theoretical Prediction

The comparison between hardened concrete test results and the theoretical prediction was carried out in this study by considering the above phenomena. Figure 13 presents the experimental and theoretical relation between compressive and split strength of OPS concrete specimens for 7- and 28-days. It should be noted that the split strength of OPS concrete can be quantified for any compressive strength data. There are a number of empirical formulas from various standards to predict the mechanical properties. The standards AS 3600, ACI 318, ACI 363R, and CEB-FIP [57–60] are recommended for lightweight concrete materials by the following Equations (9), (10), (11), and (12), respectively, to evaluate split from compressive strength.

$$f_{sp} = 0.4\sqrt{f_c} \tag{9}$$

$$f_{sp} = 0.56\sqrt{f_c} \tag{10}$$

$$f_{sp} = 0.59\sqrt{f_c} \tag{11}$$

$$f_{sp} = 0.3(f_c)^{2/3} \tag{12}$$

where $f_{sp}$ = split-tensile strength of cylindrical specimen (MPa) and $f_c$ = compressive strength (MPa).

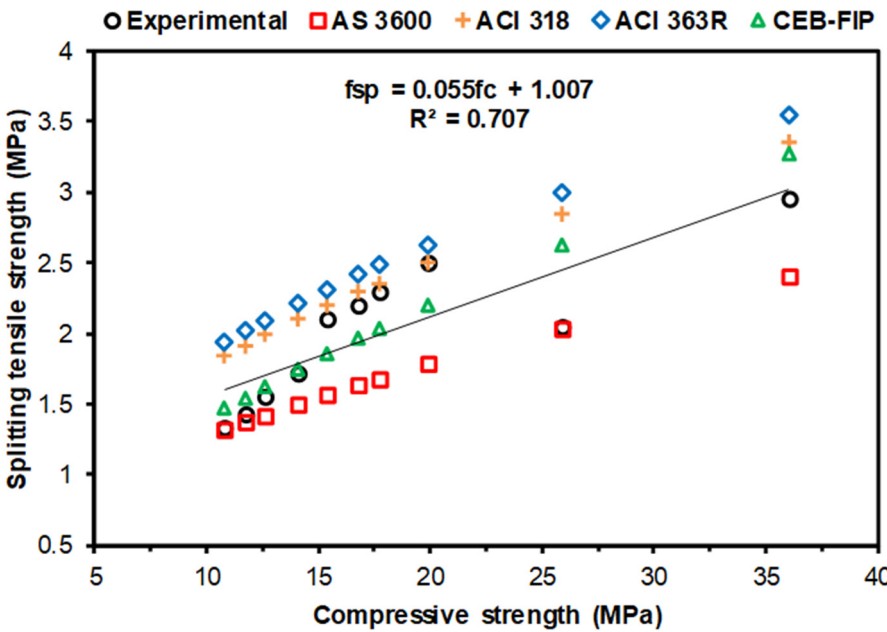

**Figure 13.** Compressive strength vs. experimental and theoretical splitting tensile strength.

It is noticed among four standards that the anticipated split strength calculated from the compressive strength of CEB-FIP predicted the closer values to the test result. In contrast, the predicted values from AS 3600 were the farthest from the test findings. In general, all the predicted data compared to the test results were within the points of all four codes of standards with the coefficient of determination ($R^2$) value of 0.707. This indicated that the current investigation prediction could accurately predict the splitting tensile and compressive strength of oil palm shell lightweight aggregate concrete. In addition, it is worth noting that the split tensile strength of palm shell aggregate lightweight concrete increased with the upsurge of compressive strength. These increments exhibited a similar estimation except for PSC-50 mix at 28 days. In their experimental study, Shafigh et al. [42] also estimated the consistent analytical relationship between the tensile strength and

compressive strength of palm shell aggregate concrete. The following correlation between compressive and splitting tensile strength of palm shell aggregate concrete can be suggested from this investigation.

$$f_{sp} = 0.055f'_c + 1.007 \tag{13}$$

The scatter plot in Figure 14 shows the compressive and modulus of rupture relationship of palm shell aggregate concrete. The flexural strength of the concrete can be predicted from the compressive strength of behavior through varying codes of standards such as AS 3600, ACI 318, ACI 363R, and CEB-FIP [57–60]. These standards were utilized to predict the flexural strength from the compressive strength properties of concrete with the following equations suggested by the code.

$$f_r = 0.6\sqrt{f_c} \tag{14}$$

$$f_{sp} = 0.62\sqrt{f_c} \tag{15}$$

$$f_{sp} = 0.94\sqrt{f_c} \tag{16}$$

$$f_{sp} = 0.46(f_c)^{2/3} \tag{17}$$

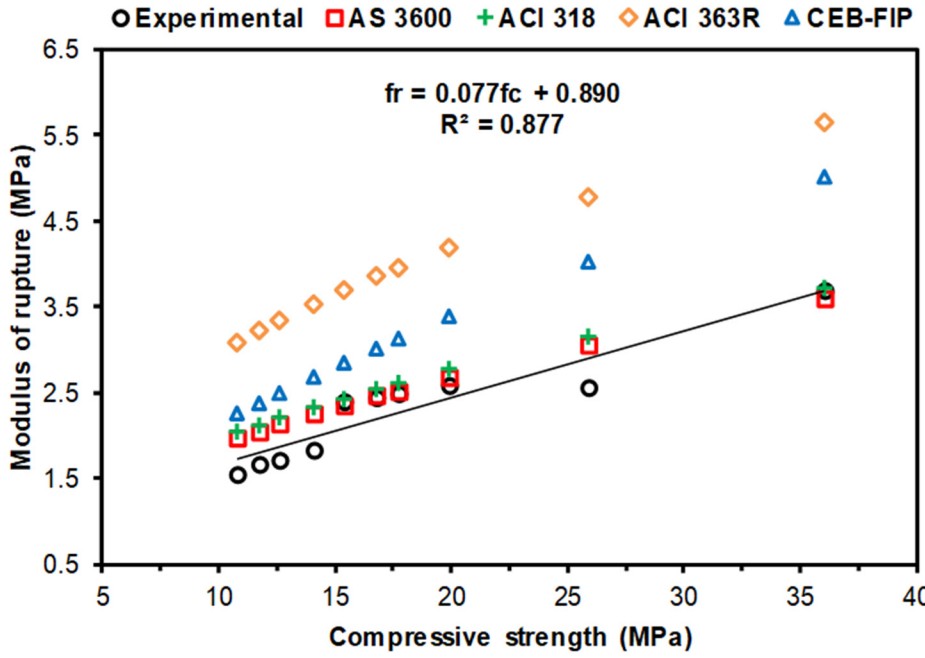

**Figure 14.** Compressive strength vs. experimental and theoretical modulus of rupture strength.

Here, $f_{sp}$ = flexural strength of concrete (MPa) and $f_c$ = cylindrical compressive strength (MPa).

It is worth mentioning that the AS 3600 exhibited the closest projected flexural strength values with respect to the corresponding compressive strength for palm shell aggregate concrete among all codes of standard equations. Consequently, the anticipated flexural strengths of ACI 363R calculated from the strength in compression were estimated at the furthest values from the current investigation. The $R^2$ value of this experimental relation was found to be about 0.877, which demonstrated that the variability of the proposed relation was small, with good reliability. The anticipation is closer compatible with the prediction by Shafigh et al. [51], where they obtained an $R^2$ value of 0.90 for palm shell aggregate concrete in their experimental investigation. However, half of the predicted

data points from the experimental results were exuded below the points of all four codes of standards.

According to the test results, Figure 15 illustrates the correlation between the flexural and split tensile strength of palm shell aggregate concrete. The $R^2$ value of this experimental result was obtained at 0.93, which indicated very little variability of the proposed relation and the best accuracy among the test values. The suggested equation of the correlation between the flexural strength of palm shell aggregate concrete and compressive strength is stated as follows:

$$f_r = 0.077 f_c + 0.890 \tag{18}$$

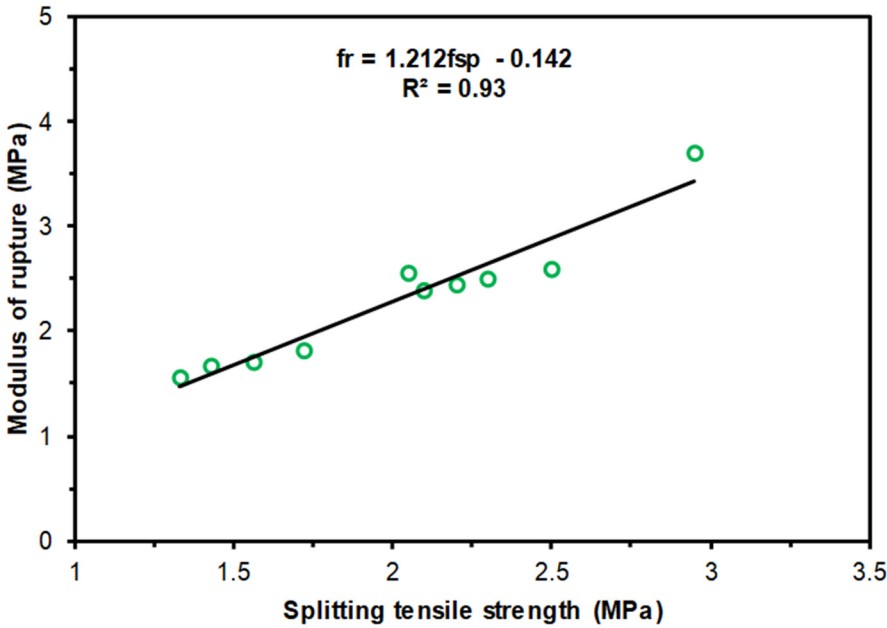

**Figure 15.** Relation between modulus of rupture strength and splitting tensile strength of OPS concrete.

Figure 16 presents the correlation between the modulus of elasticity (MOE) and the compressive strength of OPS concrete for 28 days of testing specimen. Different standards are recommended for different expressions through standard equations to anticipate the MOE of the concrete from the compressive strength. Therefore, AS 3600, ACI 318, and BS 8110 [38,57,58] standards provided the following Equations (19), (20), and (21), respectively, for predicting the MOE of the palm shell aggregate concrete based on the compressive strength.

$$E_c = 0.043 p^{1.5} \sqrt{f_c} \tag{19}$$

$$E_c = 4733 \sqrt{f_c} \tag{20}$$

$$E_c = 0.0017 w^2 f_c^{0.33} \tag{21}$$

Here, $E_c$ = modulus of elasticity of concrete (MPa) and $f_c$ = cylindrical compressive strength (MPa).

In addition, our current work successfully established a linear regression model to reflect the relationship between the experimental results of the modulus of elasticity (MOE) and the compressive strength of concrete, including palm shell aggregate. The suggested equation is as follows:

$$E_c = 0.709 f_c + 3.097 \tag{22}$$

It is observed that the anticipated values of MOE of ACI 318 code calculated based on compressive strength demonstrated the furthest prediction from the test results and code of

standard equations. In contrast, the predicted data points obtained from AS 3600 exhibited the nearest estimation with the experimental results. However, an appropriate correlation between the MOE and the uniaxial compressive strength was obtained with an $R^2$ value of 0.994 for palm shell aggregate concrete. This indicated that the variability of the proposed relationship was very small, and the reliability between the corresponding results was the best estimation.

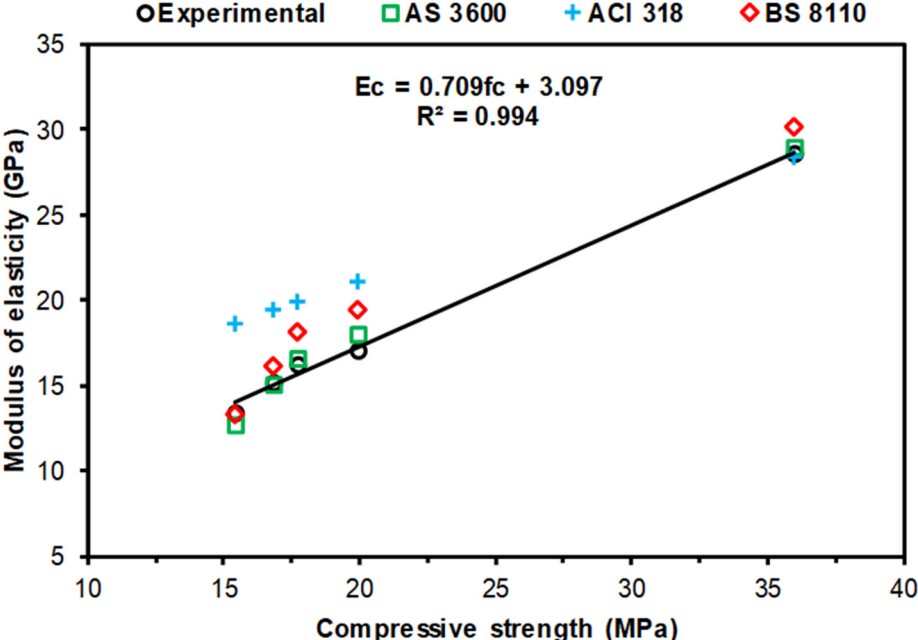

**Figure 16.** Relation between compressive strength and MOE of OPS concrete.

### 3.4. Flexural Response of OPS Beams

#### 3.4.1. Load vs. Deflection Behaviors

The load vs. deflection performance of the reinforced concrete beam without OPS aggregates and with varying concentrations of OPS aggregate are depicted for this investigation, as shown in Figure 17. The initial stiffness of the PSCB specimens is slightly increased over the conventional specimen. The material used in PSCB may exhibit this behavior to a greater extent than the material used in PSCB-0. The initial stiffness also depends on other factors, like the distribution of stress within the material and the interaction between different beam components. It is possible that the substitution rate affected these factors in such a way that the initial stiffness increased. In general, OPS-replaced beams have displayed higher deflections than the conventional coarse aggregate concrete beam; however, the maximum deflection was observed for OPS beam PSCB-50 compared to other beams. In addition, the deflection at ultimate loading was found to be 27.89 mm for control beam PSCB-0, whereas it was obtained at 37.78 mm for the beam PSCB-50. This behavior of increasing deflection of OPS concrete beams may be credited to the minor stiffness of the palm shell aggregate over the conventional coarse aggregate. Moreover, OPS is porous in nature with inferior density compared to the conventional aggregate, leading to the inferior stiffness property of the palm shell aggregate. A similar consistent finding regarding the increment of deflection of OPS beam under flexural loading was observed in the previous experimental work with varying concentrations of OPS-replaced RC beam [30]. In another work, the experimental investigation was conducted by Alengaram et al. [61], where it was reported that the OPS beam exhibited higher deflection at the ultimate loading stage with different concentrations of palm shell aggregate incorporated to prepare the RC beams.

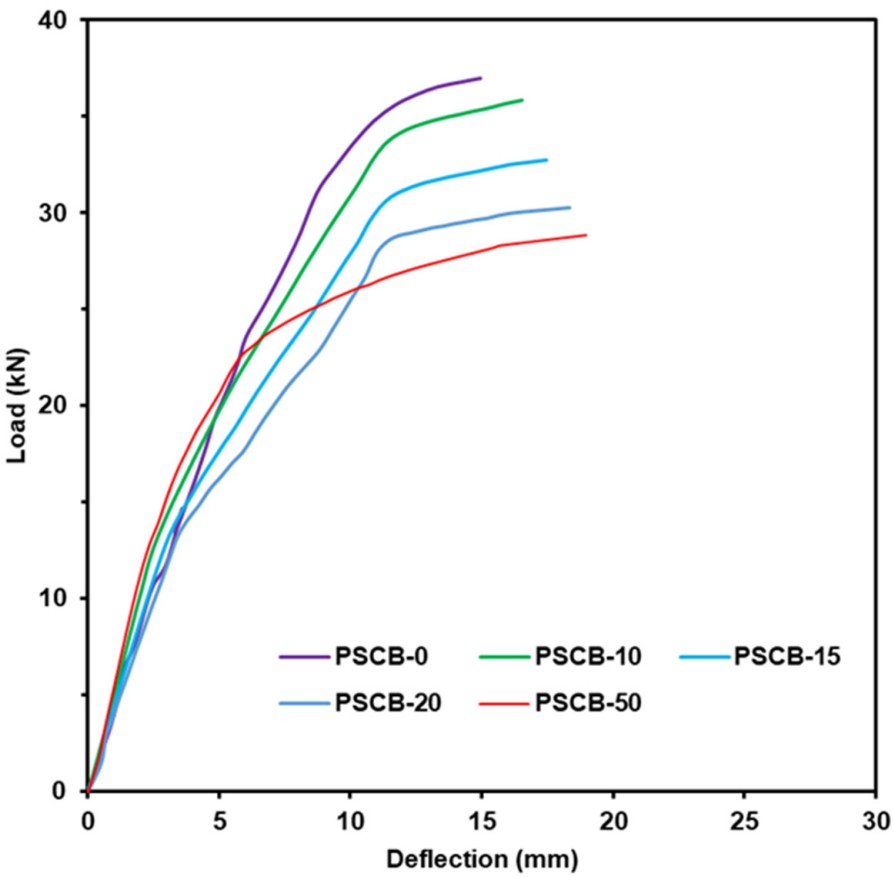

**Figure 17.** Load vs. deflection behavior of conventional and OPS aggregate concrete beams.

On the other hand, the control beam PSCB-0 demonstrated maximum flexural strength with respect to the other OPS-replaced beams. The reason for the lower ultimate load of OPS concrete beams compared to the control beam was due to the smooth peripheral surface of the OPS, more voids in the mixes, and less stiffness compared to the conventional aggregate, resulting in weak bonding between the coarse aggregate and cement matrix which prevented carrying the higher ultimate load. This is in line with the conclusion made by Olanitori and Okusami [30] in their experimental investigation, which stipulated the lesser ultimate load of the OPS beam compared to the conventional aggregate beam. The beam with the highest concentration of palm shell aggregate contained a higher steel ratio than other OPS beams; however, it exhibited a lower load capacity and the corresponding increased deflection at the ultimate stage. This may be attributed to the highest amount of OPS replacement in the beam compared to the other OPS beam, which leads to more voids in the concrete, thus weakening the bond between palm shell aggregate, cement matrix, and conventional coarse aggregate.

Further examining the test results, both without and with OPS-reinforced concrete beams at early stages showed almost linear behavior that represents the tension steel and concrete carrying force up to the yield stage of the beam as depicted in Figure 17. At the initial stage of loading, the soffit concrete of the beam can delay the tensile forces employed prior to the concrete tensile strength of the beam that has conquered the first cracking load. This behavior suggests that the beam deflection could increase sharply before the first cracking initiated on the beam surface, which was detected by visual observation and the predicted theoretical calculation. After that, this behavior is continued until the yield stage of loading, where the spalling of the compression concrete in the beam governs the steel yield. Following the yield stage, with further increase in the load, the tension soffit concrete started to exceed the neutral axis of the beam section with numerous multiple cracks that appeared in the middle third of the beam. This observation aligns with the experimental

study conducted by Alengaram et al. [55], which investigated the shear response of varying concentrations of OPS aggregate beams under flexural loading.

The percentage changes in ultimate load and deflection with yield deflection of the OPS aggregate concrete beams are depicted in Figure 18. It can also be observed from Figure 15 that the ultimate load of all the OPS beams displayed a gradually decreasing trend, albeit the corresponding deflection of the beams exhibited the increasing of the values at the ultimate stage. In addition, the yield deflection of OPS beams increases with the increase in the palm shell aggregate concentration in the mixes; in contrast, the dissimilarity of yield deflection is observed at the highest OPS-replaced beam.

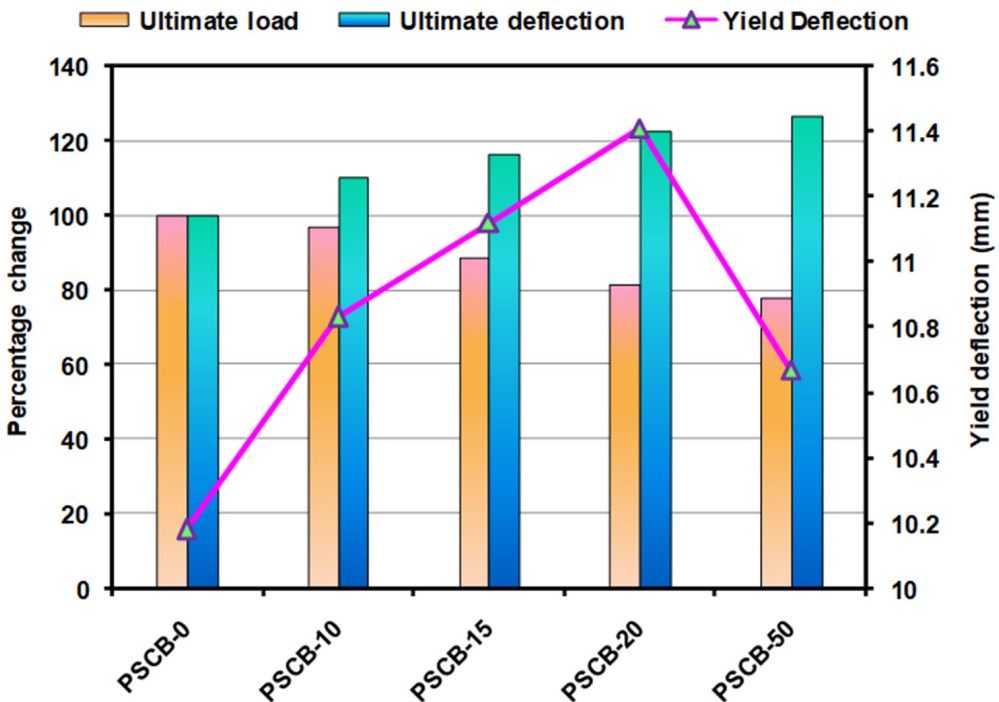

**Figure 18.** Variation of changes of ultimate loading stage with yield deformation of OPS beams.

3.4.2. Ductility Behaviors

Table 4 shows the displacement ductility of the different OPS concentration reinforced concrete beams. It is worth noting that OPS replacement beams exhibited a much more ductile nature, employing a higher ductility index than those without OPS (control) beam replacement. Other researchers also found the consistent ductility behavior of OPS beams [26,62]. It should be noted that the ductility index of palm shell concrete beam PSCB-50 was shown to be around 36% higher value than the control beam. This suggests that the OPS concrete beam will undergo considerable displacement before final failure, so appropriate measures can be taken to prevent the beam from collapsing. It also permits the concrete to undergo higher strain under increasing the beam load [48,63]. It was reported that the beams having a ductility index of 3.0–5.0 are considered adequate ductile members and could be designed against earthquake forces [64,65]. The ductility index of OPS concrete beams was obtained in the extent of 3.07–4.28, which satisfied the requirement for an RC structural member. This finding aligns with Alengaram et al. [13] for the palm kernel shell RC beams.

**Table 4.** Test and analytical results of OPS beam.

| Beam Designation | Experimental Value | | | | Theoretical Value | | Failure Mode | First Crack Width (mm) | Ductility Index |
|---|---|---|---|---|---|---|---|---|---|
| | $P_{cr}$ (kN) | $P_{ut}$ (kN) | $\delta_y$ (mm) | $\delta_u$ (mm) | $P_{cr}$ (kN) | $P_{ult}$ (kN) | | | |
| PSCB-0 | 7.6 | 37.0 | 10.18 | 14.97 | 8.0 | 38.4 | Flexural | 0.05 | 1.47 |
| PSCB-10 | 6.2 | 35.6 | 10.83 | 16.55 | 6.5 | 37.0 | Shear-tension | 0.04 | 1.53 |
| PSCB-15 | 6.0 | 32.6 | 11.12 | 17.46 | 6.2 | 34.0 | Shear-tension | 0.03 | 1.57 |
| PSCB-20 | 5.8 | 30.3 | 11.41 | 18.36 | 5.9 | 31.8 | Shear-tension | 0.02 | 1.61 |
| PSCB-50 | 7.0 | 28.5 | 10.67 | 18.95 | 7.2 | 29.4 | Shear-tension | 0.01 | 1.77 |

### 3.4.3. First Cracking and Ultimate Loading

Table 4 depicts the test results without and with different concentrations of OPS beams concerning the first cracking and ultimate loads and the observation associated with failure modes and first crack width. For the analytical investigation, the first cracking load and ultimate beam flexural strength were quantified from the rectangular stress block analysis considering the equivalent transformed cracked section according to the standard code of practice [38].

The test data emphasized that the beam without OPS concentration, referred to as the control (PSCB-0) specimen, failed through the initiation of rebar yielding followed by compression concrete spalling under the flexural loading. It can be observed that the control beam produced a first crack load of 7.6 kN in the third middle zone and then further increased the load; the tension steel bar yielded approximately the load of 32.24 kN. Following the subsequent yield stage of loading, the beam reached the ultimate position and failed due to flexure through the spalling of top fiber compression region concrete around the load of 37.0 KN. This type of failure is considered a premature failure. It also reflected the low concrete compressive strength at the compression zone. On the other hand, it is revealed that the OPS concrete beams demonstrated the lower first cracking and ultimate flexural capacities with respect to the control beam. The results strongly agree with the experimental findings by Olanitori and Okusami [30] and Alengaram et al. [61]. They observed that using various concentrations of PKS in the RC beams leads to attaining less first cracking and ultimate load-carrying capacity, especially at higher concentrations of PKS-containing beams. It is worth noting that the test results presented in Table 4 revealed an average of a 14% decline in ultimate strength in OPS aggregate beams compared to the beams without OPS.

Further comparing the experimental and theoretical data, the theoretical quantification reasonably predicts the experimental value regarding the first crack and ultimate loads of the different gradation of OPS aggregate beams. Figure 19 represents the normalized results of the theoretical and experimental load values given the first crack and ultimate loads for the varying concentrations of palm shell aggregate-replaced beams. It can be seen from Figure 16 that the theoretical first cracking and ultimate load of the OPS beam always estimated the higher load-carrying capacity with respect to the experimentally obtained load-carrying capacity of the beam. This is due to the fact that the theoretical calculation is usually predicted concerning several assumptions and also the less stiffened OPS aggregate and more voids presented in the mixes, leading to less interfacial bonding between aggregate and cement matrix that may resist achieving their full first cracking and ultimate capacities in the experimental investigation.

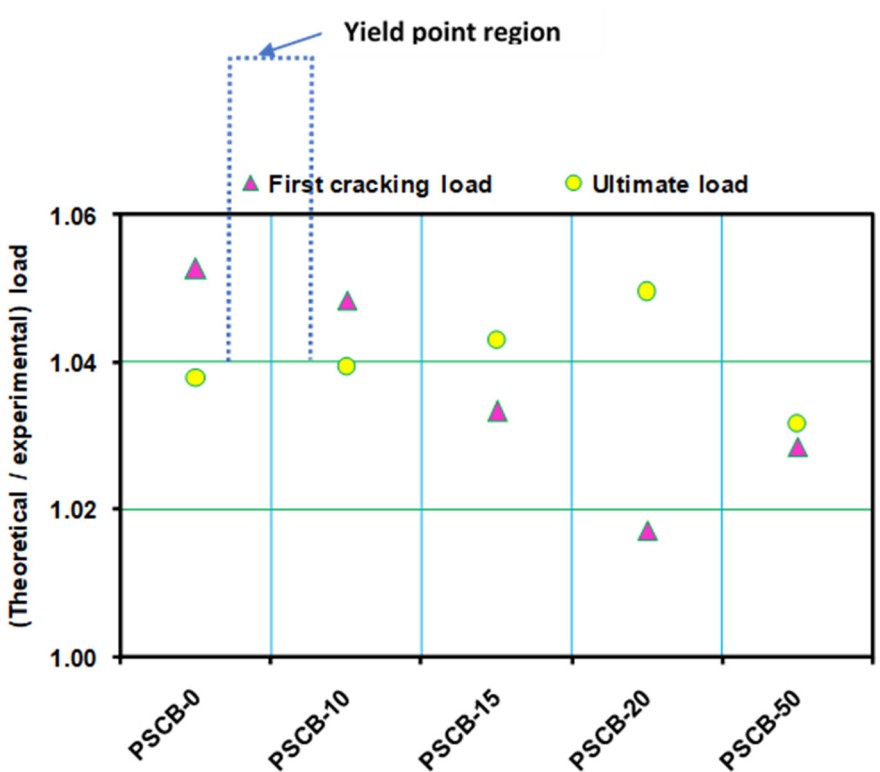

**Figure 19.** Theoretical vs. experimental loading ratio of first cracking and ultimate loads of OPS beam.

### 3.4.4. Failure Modes

It was observed from the flexural test that most of the OPS-replaced beams demonstrated a quite similar nature to the control beam at the linear elastic region. At the initial stage of load-deflection response, upon further increase of the load in the beams, when it generally exceeded the tensile stress of the concrete capacity, the small hairline cracks were visualized in the middle-third span, which is referred to as the first crack of the beam. The first crack width was measured for all the beam specimens during experimentation, and it is illustrated in Table 4. Further analyzing the failure patterns from the experimental study, it is prominent that all the varying concentrations of OPS replacement beams have reached the yield condition and failed in the same manner. It is worth noting that after the yield stage, most of the cracks produced on the tension soffit of the beam further propagated vertically towards the beam's neutral axis. The vertical nature of the cracks represents that they were recognized as flexural cracks; however, the inclined nature of the crack indicates the shear tension crack. Table 4 also designated the different failure modes of the experimentally obtained OPS beams. Afterwards, when the flexural and shear tension crack passed the neutral axis, it began to incline and continued until the final collapse of the beam. This is a predicted failure pattern of the beam under flexural loading. This behavior was observed for both the beams made with and without OPS concentration replacement in the concrete mixes. In addition, the elongated deflection due to the ultimate load of the OPS concrete beam has provided adequate warning before the final collapse of the beam, which leads to the ductile response of the sample. However, the control beam exhibited a brittle failure nature compared to the OPS beam with insufficient deformation capacity at the ultimate loading stage. The consistent failure patterns and crack propagation were also observed by other researchers at the ultimate stage of loading with sufficient ductility responses of the OPS beam under flexural test [13,31]. The typical failure patterns and crack propagation of the control beam and OPS aggregate beam are depicted in Figure 20a,b. The average crack spacing width of the control beam was determined in the range of 100–125 mm; albeit it was obtained around 50–75 mm for OPS beams that indicate the

closer and a greater number of cracks, which ultimately leads to a smaller crack opening compared to the control beam.

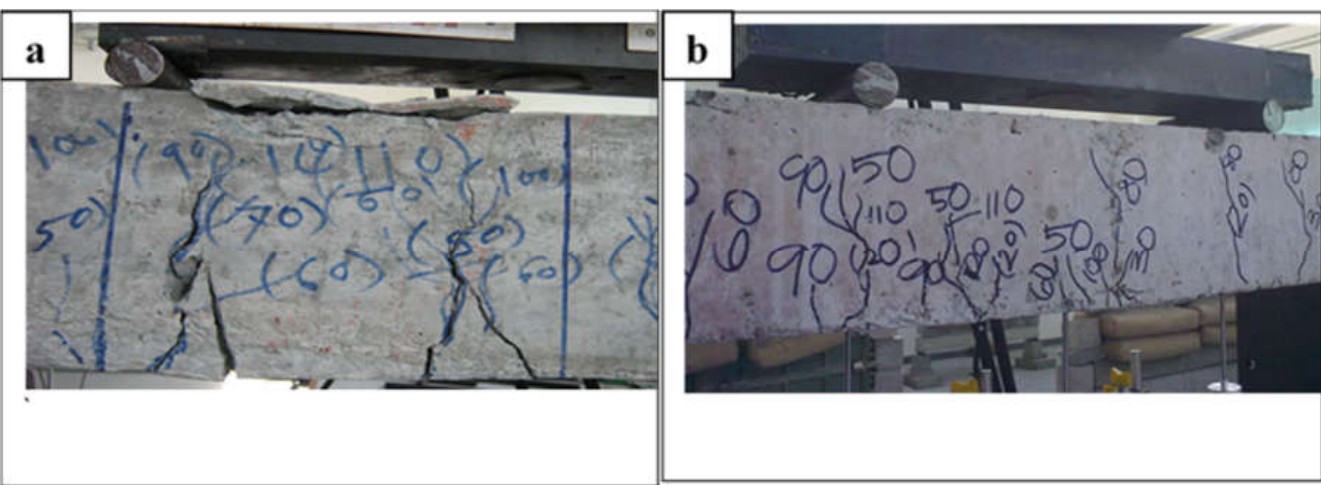

**Figure 20.** Failure patterns and crack propagation of (**a**) Control beam; (**b**) OPS replacement beam.

## 4. Conclusions

The flexural response of OPS concrete beams is experimentally investigated in this study and likened to the key mechanical properties and analytical calculations. The following conclusions could be drawn based on the present experimental results:

- All the mixes had satisfactory slump, density, and air content values that indicated the concrete medium workability and satisfied the lightweight concrete requirement. In addition, the higher concentration of OPS in the concrete mixes has decreased the slump and density values, albeit the air content of OPS concrete increased subsequently. In the slump test, the value lessened at about 27.42%, 37.10%, 43.55%, and 53.23% as compared to the control mix.
- Mix with varying concentrations of OPS replacement has shown a decreasing trend on compressive strength, splitting tensile, modulus of rupture strength, and modulus of elasticity of OPS concrete. It has been demonstrated that the compressive strength decreased 44.73%, 50.83%, 53.33%, and 57.22% at 28 days as compared to the control mix of 10%, 15%, 20%, and 50% replacement of PSC. The decrease in the splitting tensile strength of OPS concrete when increasing the concentration of OPS in the mixes is relatively smaller compared to the properties of modulus of rupture strength, modulus of elasticity, and compressive strength with respect to the conventional coarse aggregate concrete mixes.
- Analytical prediction of varying codes of practice provides a good estimation of the mechanical properties of OPS concrete in this study. In contrast, few codes of practice demonstrated a scatter prediction with the experimental results. Moreover, the analytical prediction suggests that one can easily quantify any properties from others without conducting the test. It will ultimately decrease the cost of experimentation, time, labor, and equipment utilization in the laboratory.
- The flexural responses of the OPS lightweight concrete beam resemble the behavior of the NCA concrete beam. It was obtained that although every OPS lightweight concrete beam revealed less ultimate capacity than the NCA beam, considering the material economy, the OPS lightweight concrete beams have great potential to be used in local building industries.
- The beam made with OPS aggregate concrete demonstrated a higher ductile response than the NCA concrete beam. All the OPS lightweight concrete beams showed a considerable degree of deflection, which indicated ample warning of the imminence of final failure.

- The theoretical result of varying percentages of OPS lightweight concrete beams shows close agreement with the experimental investigation. In general, it provides a good estimation of the first cracking and ultimate load-carrying capacity of palm shell aggregate lightweight concrete beams.

**Author Contributions:** Conceptualization: M.H.R.S., M.S.I. and A.S.M.A. Data curation: M.H.R.S., S.D.D., T.S.A., N.M.S.H. and M.T.H.K. Formal analysis: M.H.R.S., A.S.M.A., S.D.D., T.S.A. and M.T.H.K. Funding acquisition: M.H.R.S. and M.S.I. Investigation: M.H.R.S., M.S.I., A.S.M.A., S.D.D. and N.M.S.H. Methodology: M.H.R.S., A.S.M.A. and S.D.D. Project administration: M.H.R.S., M.S.I. and A.S.M.A. Resources: M.H.R.S., N.M.S.H., M.T.H.K. and F.S.A. Software: M.H.R.S., A.S.M.A., S.D.D., T.S.A., M.T.H.K. and F.S.A. Supervision: M.H.R.S., M.S.I. and A.S.M.A. Validation: M.H.R.S., M.S.I., S.D.D., T.S.A., N.M.S.H., M.T.H.K. and F.S.A. Visualization: M.H.R.S., S.D.D., T.S.A., N.M.S.H., M.T.H.K. and F.S.A. Writing-original draft preparation: M.H.R.S., M.S.I., A.S.M.A. and S.D.D. Writing-review and editing: M.H.R.S., M.S.I., A.S.M.A., S.D.D., T.S.A., N.M.S.H., M.T.H.K. and F.S.A. All authors have read and agreed to the published version of the manuscript.

**Funding:** This research received no external funding.

**Institutional Review Board Statement:** Not applicable.

**Informed Consent Statement:** Not applicable.

**Data Availability Statement:** Data will be available on suitable demand.

**Acknowledgments:** The authors would like to express thanks to the authority and technicians of the Heavy Structures Laboratory, Department of Civil Engineering, University Malaysia Sarawak, Malaysia for supplying support in specimen preparation, fabrication, and testing. Also, the authors would like to acknowledge the Department of Building Engineering and Construction Management, Khulna University of Engineering and Technology, Khulna-9203, Bangladesh for using the BIM laboratory to prepare the computer-aided works for the manuscript.

**Conflicts of Interest:** The authors declare no conflict of interest.

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
