# Peer review of "Mechanical Properties and Flexural Response of Palm Shell Aggregate Lightweight Reinforced Concrete Beam"

_sustainability, doi:10.3390/su152215783_

Round 1
Reviewer 1 Report
Comments and Suggestions for Authors
1.The author wrote in the Significance of this investment that this article considers beam types and different concentrations of OPS aggregates as substitutes for traditional natural aggregates to prepare RC beams and study their bending performance. However, the parameters related to beam types have not been clearly found in the entire text, and the beams have been fixed as RC beams. What is the meaning of this section, is it to study different types of beams?
2. The author is requested to provide a detailed introduction to the specific composition and loading plan of the loading device in 2.3.2 Flexural testing of beam. And from Figure 3, it can be seen that the author did not install displacement meters at the left and right ends of the beam. In the data processing, the author considered the impact of virtual displacement, as virtual displacement may have an impact on the initial stiffness, peak displacement, and ductility results.
3.The author's explanation of the mid span reflection at the yields of the bottom bar in 2.4.2 is missing characters. Please supplement.
4.The author's research on material properties in section 3.2 shows that with an increase in substitution rate of 0-50%, the compressive strength and elastic modulus both decrease significantly. In 3.4.1 Load vs. reflection behaviors, it can be clearly observed that the ultimate flexural bearing capacity decreases with an increase in substitution rate, which is consistent with the law of compressive strength. However, the initial stiffness showed an increasing trend, and the initial stiffness of PSCB-50 was obviously greater than that of PSCB-0. The author's beam structure did not change and it is still a regular RC beam. However, the law of initial stiffness increases with the increase of substitution rate. Please explain this to the author.
5. It is recommended that the author supplement the deviation and mean deviation of the experimental and calculated values of each specimen in Table 4. Please refer to https://doi.org/10.1016/j.conbuildmat.2020.121212
6.The author defined the failure of a beam as bending and shear failure in 3.4.4 Failure modes, but no obvious shear failure was found in the figure. At the same time, after calculation, the shear span ratio of the beam used in this article is 3.5, which does not match the characteristics of the diagonal tensile failure. Is the author's description of this part correct?
Comments on the Quality of English LanguageNone
Author Response
The authors appreciate the reviewer's comments, and the manuscript has been updated in response to each comment, as indicated by "yellow colored text”. The following sections present the action taken on each comment and how it has been resolved accordingly in the revised manuscript.

Reviewer 2 Report
Comments and Suggestions for Authors
In this manuscript, the results of this research are conveyed thoughtfully and completely, and they are consistent with the exceptional findings. This work is worthwhile to be publish in this journal after major revision. The following issues should be addressed:
1. The novelty needs to refinement and should be highlighted in the introduction part.
2. Abstract and conclusion not targeted; the authors should rephrase it.
3. There are some English mistakes throughout need to be checked.
4. The format of the references is not adequate and other need to be updated.
5. Maybe the author should compare their results clearly with other reported works, highlighting the advantage and disadvantages of their novel composite.
6. Introduction part, if possible, some important and relative reports should be added:
https://doi.org/10.1016/j.jobe.2022.104869
7. Error bars should be added in all figures
Author Response

(The authors gave the same response as above.)

Reviewer 3 Report
Comments and Suggestions for Authors
This study, presented for evaluation, examines the Mechanical Properties and Flexural Response of Aggregate Lightweight Reinforced Concrete Beam mechanical properties and bending response of oil palm shell obtained from palm bark wastes. Oil palm shell (OPS) was used as a natural material. The admixture consists of samples created by obtaining different percentages, and the bending of these samples is examined. This material is intended to be used as a structural and construction derivative. In this form, the study is one of the current studies in which the content of similar additives is discussed. This works topic that is currently covered. There are similar studies on the subject. There are also studies on oil palm shell. However, the subject addressed in this study is the examination of some mechanical properties of the structure obtained with natural additive materials. Although it does not fully qualify as an original study, the study contains its own experiments and findings.
It is expected that the difference from other applications will be explained in more detail in order to attract the attention of the reader. The natural additive structure as a material was examined. Bending, one of the mechanical properties of this material, is discussed. The variability of contribution rates and related changes can be considered as a contribution to the field. In the study, the additive is based on the performance measurement and comparison of the samples formed at different rates. It was determined that the bending capacity of beams at different ratios tended to decrease the strength and the results were presented graphically. The techniques and methods used in obtaining and creating materials are generally accepted scientific approaches. Performance measurements and tests are among the methods with acceptable validity. The variability of contribution rates and related changes can be considered as a contribution to the field. The results are presented as graphs and using tables. The results are the expected and scientifically valid results. It is expected that the literature part will be given weight in the study- More up-to-date sources may help to recognize the problem and explain the suggested results more clearly. In conclusion, although it is not a very innovative approach, it may interest the reader - Methods and techniques are scientific. Conclusions and comments are acceptable. The weakness of this being the case is that it can be done and its originality is questionable. References are appropriate. However, literature review needs improvement.
Comments on the Quality of English LanguageMinor editing of English language required about grammatical
Author Response

(The authors gave the same response as above.)

Reviewer 4 Report
Comments and Suggestions for Authors
Manuscript ID: sustainability-2617333 entitled "Mechanical Properties and Flexural Response of Palm Shell Aggregate Lightweight Reinforced Concrete Beam" for Journal of “Sustainability” has been reviewed.
The manuscript (review) was interesting and well-motivated. The following list of comments will help to further improve the manuscript:
+1- The most striking section of a study is the abstract section. Therefore, it must be written carefully. The “Abstract” section should be enriched with the results obtained from the study. (important results from the study should be highlighted.) (Abstract is too short.)
+2- The introduction section should contain more literature review. (2021-2023)
+3- The novelty of the study should be mentioned a little more in the introduction section.
+4- The flowchart should be added to the “Materials and Methods” section. Thus, the intelligibility of the study will increase. (use Microsoft Visio)
+5- Pictures of the all devices used in the experiments should be attached.
+6- Why were Ordinary Portland cement, river 116 washed sand, etc. materials chosen? Please explain in detail. (in “Materials and Methods” section)
+7- It would be more appropriate to give Table 1 and Table 2 as graphs.
+8- The resolution of all Figures should be increased. The thickness of the lines in the graphs should be increased.
+9- More information on tests should be given in the "Materials and Methods " section. (humidity, ambient temperature (approximately value), etc.) The dimensions of the samples should be given in 3D. How were the parameters determined? Please explain in detail. (in “Materials and Methods” section)
+10- The “Conclusions” section should be reviewed. (It should be enriched with the results obtained from study.) (Comparisons as a percentage should be added. (for example, ….. it was 25% better.)
+11- More literature studies should be added to the introduction and other sections (for example: DOI given). https://doi.org/10.1080/01694243.2023.2221391 (info about humidity, ambient temperature etc.)
***After revision, I would like to review the article again.
Comments on the Quality of English LanguageMinor editing of English language required
Author Response

(The authors gave the same response as above.)

Round 2
Reviewer 1 Report
Comments and Suggestions for Authors
The references all come from one region or country. Please add more references from other countries. such as: doi: 10.3389/fmats.2021.762568
Comments on the Quality of English LanguageNone
Author Response
Dear Reviewer,
We have cited your recommended paper as well as other countries related paper. Thank you for your time and support.
Best regards
Dr. Md. Habibur Rahman Sobuz

Reviewer 4 Report
Comments and Suggestions for Authors
Manuscript ID: sustainability-2617333 entitled "Mechanical Properties and Flexural Response of Palm Shell Aggregate Lightweight Reinforced Concrete Beam" for Journal of “Sustainability” has been reviewed.
-Abstract clearly presents objects methods and results.
“The focus of this work is to examine the mechanical characteristics and flexural response of reinforced concrete (RC) beams by incorporating oil palm shell (OPS) lightweight aggregate from oil palm shell waste. The OPS aggregates are replaced in various percentages, such as 0 to 50% of natural coarse aggregate (NCA). Mechanical properties of OPS concrete were conducted and these properties to quantify the flexural performance of RC beams. Five RC beams with several gradations of OPS aggregates were cast and tested for this investigation. The first cracking, ultimate strength, load-deflection behaviour, ductility index, and failure patterns of OPS aggregate beams were investigated as the corresponding behaviours to the NCA concrete beam. The fresh properties analysis demonstrated that lessening the slump test by varied concentrations of OPS concrete. Furthermore, compressive strength reduced 44.73%, 50.83%, 53.33%, and 57.22% compared to 10%, 15%, 20%, and 50% OPC substitution at 28 days. Increasing OPS content in concrete mixes decreased splitting ten sile strength, comparable to the compressive strength test. Modulus of rupture and modulus of elas ticity experiments exhibited a similar trend toward reduction over the whole range of OPS concentrations (0–50%) in concrete. It was revealed that the flexural capacity of beams tends to decrease the strength with the increased proportion of OPS aggregate. Moreover, crack patterns and failure modes of beams are also emphasized in this paper for the variation of OPS replacement in the NCA. The OPS aggregate RC beam's test results have great potential to be implemented in low-cost civil infrastructures.”
-Keywords are adequate.
-Scientific methods are adequately used.
-Terminology is adequate.
-Results are clearly presented.
-Conclusions are logically derived from the data presented.
“The flexural response of OPS concrete beams is experimentally investigated in this study and likened to the key mechanical properties and analytical calculations. The fol lowing conclusions could be drawn based on the present experimental results: • All the mixes had satisfactory slump, density, and air content values that indicated the concrete's medium workability and satisfied the lightweight concrete requirement. In addition, the higher concentration of OPS in the concrete mixes has decreased the slump and density values, albeit the air content of OPS concrete increased subsequently. In the slump test, the value lessened at about 27.42%, 37.10%, 43.55%, and 53.23% as compared to the control mix. • Mix with varying concentrations of OPS replacement has shown a decreasing trend on compressive strength, splitting tensile, modulus of rupture strength, and modulus of elasticity of OPS concrete. It has shown that the compressive strength decreased 44.73%, 50.83%, 53.33%, and 57.22% at 28 days as compared to the control mix of 10%, 15%, 20%, and 50% replacement of PSC. The decrease in the splitting tensile strength of OPS concrete with increasing the concentration of OPS in the mixes is relatively smaller compared to the properties of modulus of rupture strength, modulus of elasticity, and compressive strength with respect to the conventional coarse aggregate concrete mixes. • Analytical prediction of varying codes of practice provides a good estimation of the mechanical properties of OPS concrete in this study. In contrast, few codes of practice demonstrated a scatter prediction with the experimental results. Moreover, the analytical prediction suggests that one can easily quantify any properties from others without con ducting the test. It will ultimately decrease the cost of experimentation, time, labour, and equipment utilization in the laboratory. • The flexural responses of the OPS lightweight concrete beam shows resemble the behaviour of the NCA concrete beam. It was obtained that although every OPS light-789 weight concrete beam revealed less ultimate capacity as compared to that of the NCA 790 beam, considering the material economy, the OPS lightweight concrete beams have great 791 potential to be used in local building industries. • The beam made with OPS aggregate concrete demonstrated a higher ductile re-793 sponse than the NCA concrete beam. All the OPS lightweight concrete beams showed a 794 considerable degree of deflection, which indicated ample warning of the imminence of 795 final failure. 796 • The theoretical result of varying percentages of OPS lightweight concrete beams 797 shows close agreement with the experimental investigation. In general, it provides a good 798 estimation of the first cracking and ultimate load-carrying capacity of palm shell aggre-799 gate lightweight concrete beams.”
-----------------------------------------------
The authors have made significant improvements to the paper by addressing the feedback provided by the reviewers, resulting in a clearer presentation of results. Based on these revisions, the paper is now ready for acceptance.
Comments on the Quality of English LanguageMinor editing of English language required
Author Response
Dear Reviewer,
We have done minor editing of English language on the manuscript as per your suggestion with the highlighted color. Thank you for your time and support.
Best regards
Dr. Md. Habibur Rahman Sobuz

Round 3
Reviewer 1 Report
Comments and Suggestions for Authors
None